# Ion Changes and Signaling under Salt Stress in Wheat and Other Important Crops

**DOI:** 10.3390/plants13010046

**Published:** 2023-12-22

**Authors:** Sylvia Lindberg, Albert Premkumar

**Affiliations:** 1Department of Ecology, Environment and Plant Sciences, Stockholm University, SE-114 18 Stockholm, Sweden; 2Bharathiyar Group of Institutes, Guduvanchery 603202, Tamilnadu, India; premalbert2021@gmail.com

**Keywords:** cereals, chloride, cytosolic Ca^2+^, K^+^, Na^+^, pH, salt stress, signaling, wheat

## Abstract

High concentrations of sodium (Na^+^), chloride (Cl^−^), calcium (Ca^2+^), and sulphate (SO_4_^2−^) are frequently found in saline soils. Crop plants cannot successfully develop and produce because salt stress impairs the uptake of Ca^2+^, potassium (K^+^), and water into plant cells. Different intracellular and extracellular ionic concentrations change with salinity, including those of Ca^2+^, K^+^, and protons. These cations serve as stress signaling molecules in addition to being essential for ionic homeostasis and nutrition. Maintaining an appropriate K^+^:Na^+^ ratio is one crucial plant mechanism for salt tolerance, which is a complicated trait. Another important mechanism is the ability for fast extrusion of Na^+^ from the cytosol. Ca^2+^ is established as a ubiquitous secondary messenger, which transmits various stress signals into metabolic alterations that cause adaptive responses. When plants are under stress, the cytosolic-free Ca^2+^ concentration can rise to 10 times or more from its resting level of 50–100 nanomolar. Reactive oxygen species (ROS) are linked to the Ca^2+^ alterations and are produced by stress. Depending on the type, frequency, and intensity of the stress, the cytosolic Ca^2+^ signals oscillate, are transient, or persist for a longer period and exhibit specific “signatures”. Both the influx and efflux of Ca^2+^ affect the length and amplitude of the signal. According to several reports, under stress Ca^2+^ alterations can occur not only in the cytoplasm of the cell but also in the cell walls, nucleus, and other cell organelles and the Ca^2+^ waves propagate through the whole plant. Here, we will focus on how wheat and other important crops absorb Na^+^, K^+^, and Cl^−^ when plants are under salt stress, as well as how Ca^2+^, K^+^, and pH cause intracellular signaling and homeostasis. Similar mechanisms in the model plant *Arabidopsis* will also be considered. Knowledge of these processes is important for understanding how plants react to salinity stress and for the development of tolerant crops.

## 1. Introduction

Soil salinity is harmful for plant growth and development as it inhibits the uptake of essential nutrients such as K^+^ and Ca^2+^ [1]. Salinity also causes reduced water uptake, changed metabolism, ionic imbalance, and toxicity [2]. In dry and warm areas, coastal areas, and where irrigation with saline water is common, salinity is often a serious problem. Sodium (Na^+^) and chloride (Cl^−^) are the most abundant elements in saline soils and cause most harmful effects. The Earth’s crust contains 3% sodium and the seas and oceans contain more than 5% [3]. Saline soils are sometimes alkaline, which also is harmful for plants [4].

Except for halophytes and C_4_ plants, Na^+^ is not an essential element for growth or reproduction [5]. On the other hand, chloride is an essential macronutrient for many higher plants, although a high chloride concentration might have a negative impact on plant growth [6,7]. Chloride has been underestimated as a toxic element, and in some plant species it is as harmful as sodium [8]. Most plant species, which are glycophytes, cannot survive in high-salinity soils, but halophytes grow even better under salinity. High salinity causes both osmotic stress and ion toxicity, which in turn also induce oxidative stress [9,10].

Plants have several mechanisms to cope with salt stress and keep ion homeostasis under salinity. This review will focus on how important crops, especially wheat, rice, barley, pea, and beans take up sodium, chloride, and K^+^, how they sense salt stress and transfer the information by cytosolic Ca^2+^ and pH signaling to metabolic downstream reactions leading to tolerance. Some findings show that K^+^ can also take part in the signaling and will be discussed. Numerous studies are included in the present review that focus on cytosolic ion alterations of Ca^2+^, Na^+^, and pH in shoot and root protoplasts, live cells without cell walls, but also sodium, K^+^, and chloride changes in intact plants subjected to salt stress. Under stress, systemic-Ca^2+^ signaling occurs within the whole plant by Ca^2+^ waves from the local perception to distal target cells and organs. Here, we will describe the recent advances concerning systemic signaling, such as amino-acid-induced Ca^2+^ signaling.

Measurements of the influx and concentrations of different ions can be conducted by epi-fluorescence microscopy, video imaging, and fluorescent probes, which specifically bind to the ion of interest. Fluorescence tracers can also be expressed in the cells by genetic modification.

It is still unclear how the plant senses salt. According to investigations conducted on *Arabidopsis* and rice, Na^+^ should reach the cytosol prior to a further sensing process. Recent findings suggest that Na^+^ can bind to a plasma membrane sphingolipid, which then triggers a Ca^2+^ influx [11], but other possibilities for sodium sensing will also be mentioned.

## 2. Na^+^ Uptake and Accumulation under Salt Stress

### 2.1. The Hydraulic Conductivity (Lp) Affects Ion Transport

The water flow through a plant decreases under salinity and affects the ion transport. Lu and Fricke [12] investigated the root hydraulic conductivity (Lp) in wheat under different NaCl concentrations and found that the Lp of cortex cells was differently affected by NaCl concentrations under day and night, and in the main axis of roots and lateral roots. The aquaporin-inhibitor hydrogen peroxide (H_2_O_2_) reduced Lp during the night, suggesting that these proteins were important for hydraulic conductivity. The authors suggested that the changes in root Lp in response to salt stress depended on altered activity of aquaporins in root and leaves. However, aquaporins facilitate diffusion of H_2_O_2_ through cellular membranes. Therefore, it cannot be excluded that the Lp decrease is a side effect of cell membrane damage by reactive oxygen species (ROS). Aquaporins play key roles in the hydraulic regulation of other types of abiotic stress too. They are intrinsic protein channels mainly in plasma membranes, ER, vacuoles, and plastids, facilitating the diffusion of water and small neutral molecules, and dissolved gases like CO_2_ and ammonia. Interestingly, aquaporins can be regulated by signaling intermediates, cytosolic Ca^2+^, pH, and reactive oxygen species (ROS) [13]. Genetically modified aquaporins could be future candidates for improving salt tolerance in plants.

### 2.2. Uptake of Na^+^ and Cl^−^ at the Whole Plant Level

In most plant species, Na^+^ and Cl^−^ can easily be absorbed by both the main root and lateral roots. The ions are transported through root hairs into the epidermis, cortex, endodermal cells, and into the parenchyma cells, layers of pericycle, and thereafter into the xylem for further passive transport by the transpiration stream in the shoot. Solutes and water can travel by the epidermis and cortex cells into the xylem in three ways: apoplastically by the cell walls and extracellular spaces, symplastically from cell to cell by cytoplasm and plasmodesmata, openings in the cell walls, and by a transmembrane pathway by plasma membranes and cell walls [14]. The endodermal cells form a central ring structure where the radial walls are thickened (Casparin strips) with hydrophobic suberin that prevents apoplastic ion transport. In the symplastic pathway, ions have to pass through one or several membranes.

Kronzucker and Britto [15] investigated which method plants used for the uptake of ions. The reports show that monocots, like wheat and rice, more often take up Na^+^ and Cl^−^ by the apoplastic route. In rice, 50% of both Na^+^ and Cl^−^ uptakes in the shoot were apoplastic [15,16]. In *Arabidopsis* and other dicots, some part of the ion uptake was significantly apoplastic. The solute permeability coefficients in *Arabidopsis* and rice were rather similar. By the symplastic pathway, the ions have to pass through several different channels or transporter proteins.

### 2.3. Ion Uptake across a Membrane

The driving force for the movement of an ion across a membrane depends on two components: one electrical and one chemical, depending on differences in charges and ion activities across the membrane [17]. 

Since the membrane potential difference across the plasma membrane is approximately −140 mV, a positive ion like Na^+^ can easily pass into the cytosol even at low concentrations [18].

### 2.4. Cellular Uptake of Na^+^

Na^+^ is transported into a plant cell by many different channels and transport proteins, such as nonselective cation channels, NSCCs, low-affinity cation transporter, LCT1, cation-chloride cotransporters, CCCs, high affinity-K^+^ transporters, HAKs, HKT1, HKT2, and the Shaker-type K channel, AKT1.

#### 2.4.1. Cytosolic Uptake of Na^+^ in Wheat and Rice by NSCCs, CCCs, HKTs, and AKT

The cytosolic uptake of Na^+^ in wheat and rice under salinity is mainly mediated by nonselective ion channels, NSCCs, but also by transport proteins [3,15,19]. Monocots like wheat and rice use many different transporters. Davenport and Tester [20] showed that the NSCCs are the primary pathways for Na^+^ influx into wheat roots, as the influx is inhibited by Ca^2+^, a mechanism that is considered specific for that type of channel. However, later reports suggested that both K^+^ and Ca^2+^ may inhibit Na^+^ influx by NSCCs and also influx by LCT1 [15].

NSCCS are divided in depolarization-activated, hyperpolarization-activated, and voltage-insensitive channels. In barley, the voltage-insensitive NSCCs show both inward and outward currents [21]. NSCCs can also be characterized and named by their ligands and stimuli as cyclic nucleotide gated channels, CNGCs [22,23], and glutamate-like receptor proteins, GLRs [24].

Except for NSCCs, Na^+^ uptake can be mediated by cation-chloride cotransporters, CCCs, but other uptake mechanisms have been proposed as well. In wheat and rice, high-affinity K^+^ transporters, HAKs, HKT1, HKT2, low-affinity cation transporter, LCT1, and the Shaker-type K channel, AKT1, were shown to transport sodium [25,26], with references therein. In wheat, LCT1 was suggested to transport Ca^2+^ and Cd^2+^ [27]. In barley, HKTs were showed to improve salt tolerance, as overexpression caused increased tolerance [28]. The HvHKT1;1 in barley showed a higher selectivity of K^+^/Na^+^ and a high retention of K^+^ and Ca^2+^ in root cells under salinity [29].

More detailed information on Na^+^ transporters is available [15,24]. It can be concluded that, under salinity, Na^+^ is mainly transported into cereals by NSCCs, HKTs, and HAKs.

#### 2.4.2. Cytosolic Na^+^ Uptake in Arabidopsis

Results from experiments with *Arabidopsis*, a dicot, suggest that NSCCs, such as CNGCs and GLRs [3,23,30,31] but also HKTs and HAKs, can transport sodium under salt stress [24]. The aquaporin, PIP2;1, was also shown to transport Na^+^ [32].

### 2.5. Long-Distance Translocation of Sodium

#### 2.5.1. Long-Distance Translocation of Sodium in Rice by HKTs

Many reports show that HKT1 mediates Na^+^ translocation from roots to shoot in rice under salt stress. Cell-specific expression analysis by in situ PCR revealed that *HKT1* was induced in the root epidermis, vascular cylinder, and shoot mesophyll cells of the sensitive rice, indicating transport of Na^+^ from root to shoot, where it is more damaging [33]. *HKT1* induction also occurred in the shoot phloem, in the transition from phloem to mesophyll, and in the mesophyll cells of rice, suggesting a recirculating of K^+^ in the leaf. Horie et al. [34] showed that HKT1 mediates Na^+^ influx and transport, but not K^+^ influx and transport, and that HKT2 might transport both Na^+^ and K^+^. In the tolerant rice, an induction of *OsHKT2* and *OsVHA* was obtained in the root epidermis, exodermis, and xylem tissue, which indicates a K^+^ and Na^+^ uptake and transport through the root xylem [33]. This cultivar could confer salt tolerance by decreasing the expression of *HKT1* and increasing the expression of *HKT2* in the shoot, leading to a low Na^+^/K^+^ concentration ratio, which is important for salt tolerance and ionic homeostasis [35].

Investigations show that HKT1 does not transport Na^+^ from root to shoot in all species. AtHKT1 in *Arabidopsis* can prevent xylem loading and translocation to the shoot [36], but in salt-sensitive rice, Na^+^ is translocated to the shoot by OsHKT1 [33]. OsHKT1;5 is suggested to recirculate Na^+^ by the phloem to the root tissue [37,38].

#### 2.5.2. Long-Distance Translocation of Sodium in Barley by HKTs

Different transport mechanisms of HKT1;5 was demonstrated in the cultivated barley *Hordeum vulgare* and the halophyte ecotype *Hordeum marinum*, as the halophyte took up less Na^+^ into the shoot and higher K^+^ into the root, compared with *Hordeum vulgare* when subjected to 400 mM NaCl [39]. Thus, the halophyte maintained a higher K^+^/Na^+^ ratio, and also used less energy for salt tolerance than the cultivated ecotype. In several investigations, salt tolerance in the halophyte barley *Hordeum maritimum* was compared with salt tolerance in *Hordeum vulgare*. The results from those investigations are discussed in Section 4.

#### 2.5.3. Long-Distance Translocation of Sodium in *Arabidopsis* by HKTs

An investigation in *Arabidopsis* suggested that the high expression of *AtHKT1;1* in the mature part of the *Arabidopsis* root stele decreased Na^+^ accumulation into the shoot and increased salt tolerance [40]. Moreover, other research on *Arabidopsis* indicated that AtHKT1;1 controls the root accumulation of Na^+^ and retrieves Na^+^ from xylem but does not mediate root influx or recirculation in the phloem [41].

#### 2.5.4. The CCCs, Cation-Chloride Cotransporters

Loss-of-function experiments with CCCs showed that this protein can transport K^+^, Na^+^, and Cl^−^ in symport [42]. CCCs are suggested to be involved in the long-distance translocation of Na^+^ [43]. It is uncertain if the CCCs retain Na^+^ and Cl^−^ at the xylem parenchyma cells, where CCCs are expressed, or translocate these ions into the xylem. (More information on CCCs can be found in Section 4).

### 2.6. Salt Tolerance

#### 2.6.1. Plants Tolerate Salt Stress by Different Mechanisms

Salinity stress in plants causes osmotic stress, Na^+^ and Cl^+^ toxicity, and negative effects on ion homeostasis, as a high salt concentration can decrease the uptake of K^+^, Ca^2+^, and NO_3_ [26,44]. The osmotic stress is prior to the ionic stress and, in barley, more serious for the plant to combat than ionic stress, as it might be connected with the formation of ROS [45]. Osmotic stress depends on the fact that the osmotic potential is more negative outside the roots than inside under a high salt concentration, which makes water uptake more difficult.

Plants have developed several mechanisms to resist salinity [31,44]. The first minutes to days of osmotic stress cause a reduced water uptake and plant growth reduction. Plants can withstand the first phase by a reduction in cell expansion and closing their stomata; but under ionic stress, cytosolic Na^+^ and Cl^−^ concentrations can be toxic and affect the metabolism to such a degree that after a long time may cause cell death.

As reported for barley, salt tolerance does not only depend on a low uptake of Na^+^, but also on the plant’s ability to retain K^+^ and Ca^2+^ [29,46]. Plant might have a higher selectivity for K^+^ over Na^+^, by accumulating a low amount of Na^+^ in root cells, and increase their K^+^ concentration at high external Na^+^, which would lead to K^+^/Na^+^ homeostasis [44]. Higher plants can also exclude Na^+^ and Cl^−^ from the leaves, by preventing xylem loading, recirculate Na^+^ in the phloem, or transport these ions into the vacuoles to avoid toxic concentrations in the cytosol.

#### 2.6.2. Halophytes and Glycophytes

Halophytes can survive much higher concentrations of NaCl, 100–200 mM Na^+^ and Cl^−^, than glycophytes, and the tissue concentration can be much higher, >500 mM [47]. Halophytes even grow better under high salt. With some exceptions, glycophytes and halophytes might possess similar tolerance mechanisms, but the reaction strength is higher or starts more rapidly in halophytes [47,48]. Halophytes usually transport larger amounts of Na^+^ and/or K^+^ into the vacuole than the glycophytes, to use as a cheap osmoticum. Some halophytes have salt glands which can extrude Na^+^ from the leaf cells [44].

High salt conditions can cause K^+^ deficiency both in glycophytes and halophytes [49]. One reason for this might be the competition between K^+^ and Na^+^ at the uptake sites [50,51], as these ions have the same positive charges. The ionic radius is smaller for K^+^, 98 pm, than for Na^+^, 133 pm, but the hydrated Na^+^ has a larger ionic radius than hydrated K^+^. The selectivity for the uptake of K^+^ over Na^+^ is different in different plants, usually Na^+^ can inhibit the uptake of K^+^, but the opposite is less common, at least if the K^+^ concentration is less than 75 mM [52]. Another reason is that the presence of a high Na^+^ concentration causes a depolarization of the plasma membrane leading to K^+^ efflux from the cells by GORK and NSCCS channels [53] (Figure 1).

Depolarization also causes an influx of Ca^2+^ [54]. The depolarization is not always negative as it may lead to an activation of the H^+^ATPase, and the protons pumped out to apoplast can be used by the SOS1 antiporter. A more recent report suggests that GORK, an outward-rectifying K^+^ channel, may operate to switch off energy-consuming anabolic reactions and instead use the energy for stress adaptation [55]. The K^+^ efflux could be a signaling mechanism under salt stress; for instance, an increased H^+^ATPase activity and K^+^ influx from the vacuole to compensate for the K^+^ loss. The SKOR channel, STELAR K^+^-outward rectifier, might also be involved in K^+^ efflux induced by depolarization.

#### 2.6.3. Regulation of Na^+^ Transport at the at Xylem/Parenchyma Cell Border

Not only is a low cytosolic Na^+^ in the plant cells important for tolerance, but also in the prevention of ion loading into the xylem at the parenchyma–xylem border for further transport to the shoot. It was shown that both the root and shoot of the halophyte quinoa had a higher K^+^ concentration than the root and shoot of the glycophyte pea, and under salinity this concentration was even higher [56]. Despite a higher concentration of Na^+^ in quinoa roots than in pea roots, the K^+^/Na^+^ ratio was higher in quinoa.

To explain these results, electrophysiological measurements of K^+^ and H^+^ fluxes from mechanically isolated root xylem of pea and quinoa were performed [57,58]. The addition of 20 mM NaCl and ABA to the xylem medium, mimicking the natural xylem solution [59], caused a strong K^+^ efflux from the stelar cells of pea but no efflux from quinoa [56], reflecting K^+^ retention in quinoa roots. In pea, K^+^ was translocated to the shoot to compensate for a lower uptake of K^+^ in the root. There was also a H^+^ efflux from the stelar tissue of both species, but the efflux was more pronounced in pea. ABA accumulates in the root under salt stress [60]. There it might stimulate the SOS protein [61]. The addition of 50 µM ABA induced a net H^+^ uptake into xylem-parenchyma cells of both species and a net K^+^ efflux, suggesting an ion exchanger at the xylem–parenchyma interface [62]. The results corroborate results showing a higher concentration of K^+^ in the shoots of quinoa than in pea under salt stress [56]. Quinoa keeps more K^+^ as osmotic regulation in the roots than pea does. The high concentration of Na^+^ found in the shoot of quinoa may depend on the fact that quinoa uses Na^+^ as osmoregulation in the shoot.

#### 2.6.4. Different Barley Cultivars Differ in Salt Tolerance

Metabolomic and transcriptomic analyses of barley genotypes showed that the halophyte *Hordeum marinum* under salt stress used more Na^+^ and K^+^ ions for osmotic regulation and root tolerance than the cultivated *Hordeum vulgare*, and also increased the glycolysis and TCA cycle to obtain a high energy supply, necessary for shoot tolerance [63].

#### 2.6.5. Tolerant Rice Cultivars Have Different Salt-Tolerance Mechanisms

In three rice cultivars showing different salt tolerance, the most sensitive cultivar, cv. VD20, accumulated less Na^+^ than the other two cultivars [64]. The most tolerant cv. AGPPS114 accumulated more Na^+^ and also contained higher concentrations of proline and glycine betaine as osmotic regulation than the other two cultivars. Moreover, VD20 showed a higher expression of the HKTs transporters, HKT1;4 and HKT1;5, than the other cultivars, as analyzed by real-time PCR. The authors reported that the tolerant cultivar showed a higher expression of the *SOS1* and *NHX1* than the sensitive cultivars, which might explain the salt tolerance of the former cultivar.

#### 2.6.6. SOS1 Role in Salt Tolerance

A low concentration of Na^+^ in the cytosol is important for salt tolerance. Plants are able to transport Na^+^ from the cytosol into apoplasts by the Na^+^/H^+^ antiporter SOS1. The localization of SOS1 in the plasma membrane was revealed by confocal imaging of a SOS1–green fluorescent protein fusion in transgenic *Arabidopsis* [61]. Expression analyses showed that SOS1 was present in the plasma membranes of root tips and in parenchyma cells of the xylem/symport boundary. Thus, the SOS1 protein should have another function too: to reduce the transport of Na^+^ from xylem parenchyma cells into the xylem vessels, which would also be of importance for salt tolerance. Conflicting results were recently published from an investigation on the gene expression of *SOS1* and Na^+^-flux measurements, which stated that it is only the SOS1 transporters in the outer root tissue that exclude Na^+^ from the root cytosol, but that SOS1 operating in the stele actively loads Na^+^ to the xylem transpiration stream [65].

Cytosolic Na^+^ concentration and fluxes in living cells can be analyzed by dual-wavelength fluorescence microscopy and the sodium-binding fluorescent dye SBFI, AM [66]. By the use of this technique, measurements on mesophyll protoplasts from *Arabidopsis* shoots showed that the sos1 mutant took up more Na^+^ into the cytosol than did the Wt protoplasts when the external solution contained 100 mM NaCl [67]. Moreover, the *Arabidopsis* nhx mutant, localized in tonoplasts and having a functional SOS1, also took up more Na^+^ into cytosol than the Wt, probably because both the Na^+/^H antiporters are involved in the efflux of Na^+^ from cytosol. The main function of NHX is believed to be a regulator of pH and intracellular homeostasis [68]. In *Arabidopsis*, there are four isoforms of NHX and nhx triple and quadruple knockouts showed reduced growth. A lack of any vacuolar-NHX activity resulted in reduced Na^+^ uptake and no K^+^ uptake, suggesting that these antiporters take up K^+^ but also some Na^+^. They also reported a Na^+^-uptake transporter, which was independent of proton transport.

#### 2.6.7. Na^+^ and K^+^ Transport into the Vacuole

For a long time, it has been stressed that the Na^+^/H^+^ antiporter NHX in the tonoplast and in some endomembranes is important for salt tolerance, as it might transport Na^+^ out from the cytosol. However, the work in [69] reported that K^+^ concentration decreased when the Na^+^ concentration was increased in the vacuole. Moreover, coordinated transport by NHX and KEAs, K^+^-efflux antiporters, was reported to result in salt tolerance [70]. Thus, NHX might be more important for K^+^ transport into the vacuole and for pH regulation, even if NHX to a lesser degree also mediates the transport of Na^+^ [68]. Recent findings suggest that the CCX, a cation/Ca^2+^ exchanger, is suggested as a better candidate for salt tolerance than NHX, as it deceases high concentrations of Na^+^ in cytosol and reduces reactive oxygen species [71].

### 2.7. Measurements of Cytosolic Ion Changes in Different Species/Cultivars under Salinity

To compare the influx kinetics of Na^+^ species, D’Onofrio and coworkers (2005) [72], Kader and Lindberg (2005) [73] (Figure 2), and Sun et al. [56] used epifluorescence microscopy and the Na^+^-binding fluorescent dye SBFI, AM, the acetoxymethyl ester of the benzofuran isophtalate, SBFI. The fluorescent Na^+^ indicator CoroNA Green, AM, can also be used to measure Na^+^ concentrations in the cytosol. Dyes in AM-ester form can penetrate the plasma membrane and enter into the cytosol, where they are split by esterases into the Na^+^-binding fluorescent form. Ratiometric measurements at two excitation wavelengths make the result more reliable than one-wavelength measurement, as different dye concentrations, photobleaching, and the thickness of the cell show little effect [66,74].

#### 2.7.1. Cytosolic Na^+^ Influx and Efflux from Salt-Tolerant and -Sensitive Species of Quince, Sugar Beet, and Wheat Differ

D’Onofrio et al. [72] showed that the amplitude and duration of Na^+^ influx into protoplasts from species that are highly salt tolerant, such as quince (*Cydonia oblonga* Mill), salt tolerant sugar beet (*Beta vulgaris* L.cv. Monohill), and less tolerant wheat (*Triticum aestivum* L. cv. Kadett) were in the order: wheat > sugar beet > quince. The quince protoplasts took up sodium only from a Ca^2+^ free buffer containing 200–400 mM NaCl. The Na^+^ influx was transient in quince, but in wheat and sugar beet it increased to a certain level and was then stable. As 1.0 mM external Ca^2+^ inhibited the influx, it is likely that Na^+^ influx at high salt was mediated by nonselective-cation channels [30,75]. Only the halophyte quince was able to carry out a rapid efflux of Na^+^ from the cytosol.

#### 2.7.2. Cytosolic Na^+^ and pH Changes Are Different in the Halophyte Quinoa and the Glycophyte Pea

Different dynamics were obtained in the cytosolic influx of Na^+^ in the glycophyte pea, Pisum sativum, and the halophyte quinoa, *Chenopodium quinoa* [56]. The cytosolic Na^+^ concentration, [Na^+^_cyt_], was analyzed in mesophyll protoplasts after the cultivation of seedlings with and without 100 mM NaCl, and upon addition of NaCl to the protoplast medium.

The addition of 100 mM NaCl to control protoplasts of quinoa caused a transient [Na^+^ _cyt_] influx and the maximal concentration was obtained after 360 s, at the same time as for salt-adapted quince [56,72]; but when NaCl was added to salt-cultivated quinoa protoplasts, the maximal Na^+^ influx was obtained later (after 450–500 s) and was mainly transient [56].

The addition of NaCl (50 mM) to pea control protoplasts caused another reaction: the influx increased and was then stable at the same Na^+^ level. In addition, NaCl addition to pea protoplasts from seedlings pretreated with salinity showed a gradual increase for a long time.

The different reaction obtained in salinity-cultivated quinoa suggests an adaptation mechanism and might depend on a rapid activation of the H^+^-ATPase, which was absent in pea, as the Na^+^ influx in pea increased with time [76]. Quinoa grows optimally at 100–200 mM NaCl and is tolerant not only to salt stress but also to drought and frost, and thus is suitable for cultivation in areas where no other plants survive [77].

#### 2.7.3. Cytosolic Na^+^ Influx and Efflux in Tolerant and Sensitive Rice

The same differences in Na^+^-influx dynamics were also obtained when comparing the influx in tolerant rice cv. Pokkali with sensitive rice cv. BRRIDhan29 [73].

Pharmacological analysis indicated that NSCCs were the main pathways for Na^+^ influx in the tolerant rice cultivar, but that both NSCCs and high-affinity K^+^ channels HKTs contributed to influx in the sensitive one. The transient influx of Na^+^ suggests that tolerant rice has a mechanism for fast extrusion of Na^+^ that protects the cytosol from ion toxicity. This rice cultivar also has less PM permeability to Na^+^ compared to salt-sensitive rice [78]. Inhibitor analysis suggested that tolerant rice transported Na^+^ from the cytosol into the vacuole by the Na^+^/H^+^ antiporter NHX in the tonoplast, but the sensitive cultivar transported Na^+^ out of the protoplast by the Na^+^/H^+^ antiporter SOS1 in the plasma membrane [79,80,81,82].

Expression analyses by real-time RT-PCR of the *OsVHA* genes, which encode the H^+^-ATPase in the tonoplast demonstrated that the *OsVHA* transcripts were induced immediately after Na^+^ stress in the tolerant rice cultivar, but in the sensitive one, the expression of *OsVHA* was low and delayed 6 h. The tonoplast H^+^-ATPase is supposed to be important for salinity tolerance as it builds up an electrochemical gradient for the transport of cations into the vacuole [34,83]. These findings confirmed the results from protoplast experiments that showed a fast efflux of sodium from the cytosol of the tolerant rice, but little sodium efflux from the sensitive rice [33,73].

## 3. Cl^−^ Uptake and Transport under Salinity

Cl^−^ is primarily present in the soil solution as the chloride monovalent anion, and unlike other important soil anions like nitrate and sulphate, Cl^−^ is not chemically changed by soil bacteria. Under salt (NaCl) stress, Cl^−^ passively enters root cells and radially enters into xylem vessels for translocation to the shoot. At a low concentration, Cl^−^ is taken up in the root by Cl^−^/2H^+^ symporters in a secondary active way. At high concentrations in the cytoplasm, both Na^+^ and Cl^−^ ions are biologically toxic to plants [84]. Reports show that in glycophytic species, the root-Cl^−^ efflux is strongly associated with shoot Cl^−^ exclusion and salt tolerance [85,86,87]. A low concentration of Cl^−^ in the roots or shoots of several plants, such as faba bean, barley, Lotus, and Chrysanthemum, is positively correlated with salt tolerance [8,88,89]. The salt tolerance of many plants is linked to an effective management of Cl^−^ uptake and transport. Teakle and Tyerman [84] highlighted key Cl^−^-transport characteristics related to plant salt tolerance: (i) decreased net Cl^−^ uptake by roots, (ii) decreased net xylem Cl^−^ loading, (iii) intercellular Cl^−^ compartmentation, (iv) intracellular Cl^−^ compartmentation, and (v) phloem recirculation and translocation within the plant. Effective Cl^−^ exclusion from roots or shoots could prevent excessive Cl^−^ accumulation in plant tissues. In Figure 3, the overall mechanisms of Cl^−^ transport are briefly outlined.

### 3.1. Antagonism between Cl^−^ and Other Anions in Wheat and Other Cereals, and in Tomatoes and Rose Plants

According to Abdelgadir et al. [90], there is an antagonistic relationship between Cl^−^ and nitrate (NO_3_^−^) when external Cl^−^ concentrations are too high, which inhibits wheat development and yield [91]. Interestingly, phosphate (PO_4_^3−^) and Cl^−^ have also been reported to face close competition for anion–anion absorption. Such competition has also been documented for tomatoes and rose plants, in addition to cereals [92]. It appears that Cl^−^ tends to impede crop growth and development under NaCl salinity by causing phosphorus and sulphur deficiencies by blocking PO_4_^3−^ and SO_4_^2−^ absorption. This can occur through two different mechanisms: either Cl^−^ leaks from protein pores, which quantitatively dislocate SO_4_^2−^ or PO_4_^3−^ and cause a dramatic drop in their uptake, or antagonistic competition for a binding site at transport proteins of salt ions between Cl^−^ and counter-cations.

### 3.2. Wheat Leaves Might Accumulate Less Na^+^ and Cl^−^ Than Leaves of Barley, Canola, and Chickpea

Under salinity, wheat significantly suffers from poor plant development and water uptake, but its leaves exhibit far lower levels of Na^+^ and Cl^−^ than those of barley and canola [93]. The Cl^−^ content in wheat leaves increased to 84% with a rise in subsoil NaCl from −126 to −270 osmotic potential values (ECe values from 3.5 to 7.5 dS/m). This study demonstrates that at all subsoil salinity levels, wheat has a superior salt-exclusion mechanism at the roots than canola or chickpea. It was hypothesized that there was limited transfer of Na^+^ and Cl^−^ from roots to shoots in wheat plants, since the concentration of Na^+^ and Cl^−^ in wheat leaves was much lower than that of barley, canola, and chickpea leaves at the highest level of subsoil NaCl [93].

### 3.3. Chloride Channels and Transporters under Salinity

Inadequate compartmentalization of Cl^−^ between the apoplasm, the cytoplasm, and the vacuole in the leaves might cause salt toxicity [84]. In root cells, Cl^−^ is transported inward via Cl^−^/H^+^ co-transporters, anion channels, and nitrate transporters (NRTs). Different anion channels are involved in the passive transfer of Cl^−^ during salinity. Under salt stress conditions, the rate-limiting step for Cl^−^ accumulation in the shoot is the root-to-shoot xylem transport pathway [94].

#### 3.3.1. Voltage-Dependent Chloride Channels CLCs

Prokaryotic and eukaryotic organisms contain voltage-gated chloride channels of the CLC-type family, which mediate passive Cl^−^ transport driven by the electrochemical gradient. Under salt stress, these channels mediate Cl^−^ homeostasis in plants.

The regulation of net Cl^−^ absorption under salinity depends on anion channels that open to allow anion outflow. In barley plants pre-treated with 200 mM NaCl, Yamashita et al. [95] discovered an increase in Cl^−^ permeability of plasma membrane vesicles from the roots. Various investigations have reported the presence of Cl^−^ efflux channels in root plasma membranes [96,97,98,99]. Electrophysiological studies found that the plasma membrane of *A. thaliana* cortical cells and epidermal root hairs exhibited substantial levels of depolarization-activated anion efflux channels [100,101].

##### Uptake of Na^+^ and Cl^−^ and Their Translocation from Root to Shoot May Differ in Different Wheat Species and Cultivars

Two wheat species, *Triticum aestivum*, cv. Tanit, and *T. durum*, cv., were compared by Ouergh et al. [102]. The cultivar Ben Bachir showed a noticeable difference in the ion partitioning between the roots and the shoots. Tanit roots and Ben Bachir shoots were the primary locations where Na^+^ and Cl^−^ ions accumulated specifically. Compared to Ben Bachir wheat plants, Tanit plants accumulated significantly less Na^+^ and Cl^−^ ions in their leaves. Ben Bachir leaves accumulated more Na^+^ and Cl^−^ ions under salinity, and this was accompanied by a shift in the osmotic potential values to more negative values. Ben Bachir demonstrated traits of NaCl-tolerant species. Furthermore, the photosynthetic properties of the resistant Ben Bachir cultivar were unaffected by salinity stress.

Cl^−^ uptake and transport rates may vary among genotypes of wheat and increase linearly with increasing salt. There is no connection between the accumulation of Na^+^ and Cl^−^ ions or the total of Na^+^ and K^+^. Rates of Cl transport from roots to shoots in all genotypes were increased with increasing salinity. The pattern of root Cl^−^ concentration was identical to that of Na^+^ [103].

Experiments with wheat protoplasts from the epidermis and cortical cells demonstrated the presence of outward-rectifying anion channels that are triggered by depolarization (OR-DAACs). While the characteristics of these channels are similar across species, the rates of activation and rectification vary [104,105].

##### Cl^−^ Influx Channels in Rice by CLCs

Rice OsCLC1, a voltage-dependent Cl^−^ channel, demonstrates an adaptational response to elevated Cl^−^ but its involvement in nitrate absorption is less probable in highly salinized environments. OsCLC1 coordinates the control of anion and cation homeostasis in salt-treated rice, contributing to salinity adaptation. OsCLC1 transcript levels were downregulated in the leaves and roots of the salt-sensitive Cl^−^ accumulating rice line IR29 in response to salt stress, but expression was briefly upregulated in the salt-tolerant Cl^−^ excluding rice line Pokkali [106]. OsCLC-1 and OsCLC-2 are homologs of the tobacco CLC-Nt1. OsCLC-2 was exclusively expressed in the roots, nodes, internodes, and leaf sheaths, while OsCLC-1 was expressed in the majority of tissues. Treatment with NaCl enhanced the level of OsCLC-1 expression but not OsCLC-2 [107].

##### CLC-Channels in Tonoplast of Soybean and Cotton Decreases Cl^−^/NO_3_^−^ Ratio

Salinity caused tonoplast-localized GsCLC-c2 in soybean to transport Cl^−^ more efficiently than NO_3_^−^ into the vacuoles in a pH-independent way. Therefore, GsCLC-c2 could lower the Cl/NO_3_ ratio by maintaining a relatively high NO_3_ level in the aerial parts of salt-stressed plants. Thus, is likely that GsCLC-c2 reduces salt stress in plants by preventing excess Cl^−^ within the vacuoles of root cells and preventing Cl^−^ from translocation to shoots [108].

Cotton plants subjected to chloride concentrations (by NaCl or KCl) activate the tonoplast-chloride channel GhCLCg-1. A silencing of GhCLCg-1 increased the Cl^−^ concentrations in the roots, stems, and leaves and the Na^+^/K^+^ ratio in the stems and leaves, resulting in reduced salt tolerance. Thus, GhCLCg-1 can positively control salt tolerance by regulating ion accumulation of upland cotton [109]. *GhCLCg-1* transcript levels were greater in roots and leaves during salt stress but not in stems. By increasing the concentration of Cl^−^ in the roots and reducing the quantity of Cl^−^ delivered to the shoots, *Gossypium hirsutum* ghclg1 and *A. thaliana atclcg* mutants altered plant sensitivity to chloride and elevated Cl^−^ concentrations [110,111].

##### Voltage-Dependent Influx Channels in Barley and Maize

Patch-clamp studies with protoplasts of barley [112] and maize [113] root xylem parenchyma cells identified multiple Cl^−^-permeable anion channels, including an inwardly-rectifying anion channel (X-IRAC), activated by plasma-membrane hyperpolarization, a slowly-activating anion channel (X-SLAC), and a quickly-activating anion channel (X-QUAC).

##### Voltage-Dependent SLAC/SLAH Channels: *Arabidopsis*

The slowly-activating (S-type) channel called slow-type anion channel associated/SLAC1 is activated by depolarization and is controlled by the external anion activity [114,115,116]. SLAC1 mediates the efflux of chloride or nitrate. In xylem-pericycle cells, the anion channels SLAH1 and SLAH3 of *Arabidopsis* co-localize. The heteromeric channels function jointly to form chloride-conducting channels. Soil salinity reduces *SLAH1* expression, which results in Cl^−^ exclusion from the shoot. *AtSLAH1* expression significantly decreased when the soil was highly salinized. AtSLAH1 loss-of-function mutants only had a xylem-sap chloride concentration that was half that of the wild-type. Under salt stress, plants modify the assembly and differential expression of the SLAH1/SLAH3 anion-channel subunits to regulate the distribution of NO_3_^−^ and Cl^−^ between the root and shoot. Without considerably influencing the inclusion of NO_3_^−^, SLAH1 abundance contributes significantly to the Cl^−^ exclusion [117].

##### The Involvement of SLAC/SLAH in Chloride Efflux from Barley and *Arabidopsis*

The protein family known as homologues (SLAC/SLAH) interacts with K^+^-channels like KAT1 and AKT2 and regulates K^+^ homeostasis in plants [118,119]. According to Planes et al. [120], *A. thaliana* root-epidermal cells emit Cl^−^ when exposed to ABA. Microelectrode measurements of SLAH3-dependent Cl^−^ currents in root epidermal cells suggest that S-type channels may play a role in controlling net Cl^−^ uptake in plants. Salinity-sensitive barley cultivars had more Cl^−^ in the root cortical cells than salt-tolerant types had [121]. It was shown that transcripts of the anion channels *HvSLAH1* and *HvSLAC1*, which facilitate Cl^−^ efflux, were positively correlated with grain yield in the barley leaves and this shows that in barley, shoot Cl^−^ exclusion and salt tolerance are related [122]. Furthermore, the downregulation of SLAH1 by salt stress suggests that the expression of SLAH3 and SLAH1, which is linked to xylem Cl^−^ loading, may play a role in salt tolerance [123].

##### Ca^2+^-Activated Cl^−^ Efflux Channel in *Sorghum*

It was observed that a voltage-dependent anion channel in mesophyll cells, activated by an increase in cytosolic Ca^2+^, mediated chloride-ion efflux depending on the extremely negative membrane potential, producing an outward-directed anion gradient. Cl^−^ was shown to be extruded from the roots of salt-stressed sorghum plants [124].

##### Intracellular Chloride-Channels in *Arabidopsis*

In the epidermal cells of *Arabidopsis* hypocotyls, a voltage-dependent anion channel with two functional modes: rapid and slow modes was discovered [125]. Two different forms of cytosol-negative potential-activated (hyperpolarization-activated) anion channels, VCl for Cl^−^ and VMal for malate, are present in the tonoplast [126]. In electrophysiological studies using planar lipid bilayers, VCCN1 was shown to perform as a voltage-dependent Cl^−^ channel, and it was consequently postulated that it mediates Cl^−^ import into the thylakoid lumen of *Arabidopsis thaliana* [127].

##### Chloride Efflux Channels in Guard Cells

A guard cell anion channel (GCACl) was observed in the plasma membrane of isolated faba bean protoplasts [128]. The control of salt efflux from guard cells by voltage-dependent anion channels was validated by Cl^−^ currents of individual guard cells obtained in patch-clamp experiments. The sensitivity of GCACl to chloride is within the physiological range for the number of exogenous anions [129].

#### 3.3.2. Stretch-Activated Cl^−^ Efflux Channels in Guard Cells and Pollen Tubes

Along with voltage-gated channels, cells of higher plants have stretch-activated channels in the plasma membrane. The existence of stretch-activated Cl^−^, Ca^2+^, and K^+^ channels in the plasma membrane of guard cells was demonstrated by Cosgrove and Hedrich [130]. Due to the fact that stomatal guard cells have a variety of mechano-sensitive ion channels, they may be able to translate a change in salinity-induced xylem pressure into a change in stomatal apertures [131,132]. MSL10 (mechanosensitive channel-like 10) prefers Cl^−^ over Na^+^ (6:1) and may be involved in Cl^−^ efflux to release membrane tension after the channel opens [133]. Pollen tubes from mutant MSL8 plants rupture more frequently, which is consistent with the observed efflux of Cl^−^ during pollen tube formation and suggests that MSL8 is involved in maintaining cell osmolarity [134,135].

#### 3.3.3. Ion Channels in Xylem Parenchyma Cells of the Root

To release ions into the xylem, anion channels in the xylem parenchyma of roots may work similarly to those in guard cells [97]. Activation of the KORC (K^+^ outward rectifying conductance) channel, which was found in both the xylem parenchyma cells of barley and stelar cells of maize, causes Cl^−^ efflux via this channel. The downhill gradient and the negative membrane potential of the plasma membrane induce anion efflux from the cytoplasm into the extracellular space. This leads to a depolarization, which may facilitate a propagation of signals in the cytoplasm by the plasmodesmata connections between cells [131].

#### 3.3.4. ALMT Channels

R-type anion proteins are encoded by the aluminum-activated malate channel ALMT family in plants. It has been suggested that the maize plasma-membrane channel ZmALMT1, which is selective for sulphate, Cl^−^, and NO_3_^−^, is mostly expressed in mature root tissues [136]. The anion channels ALMT6 and ALMT9 are permeable to a variety of anions, including Cl and malate. Smaller stomatal openings in AtALMT9 plants make them more resistant to osmotic stress, and AtALMT9 also mediates inward-chloride currents that are activated by the presence of low amounts of cytoplasmic malate [136]. It is known that the tonoplast transporters/channels CLCa, CLCc, ALMT9, and ALMT6 all mediate anion inflow into the vacuole, suggesting that they may all work similarly to DTX33 and DTX35 (detoxifying efflux carrier 33 and 35).

### 3.4. Ca^2+^, Boron, Malate, and Aluminum Affect the Accumulation of Cl^−^ in Wheat

Every wheat genotype under investigation showed a notable capacity to regulate the accumulation of Cl^−^. As the Ca^2+^ concentration rose, so did the Cl^−^ concentration. This may be related to the rise in K^+^ that occurs when Ca^2+^ increases. Furthermore, it was discovered that no variations in Cl^−^ concentration were seen between many genotypes, including tetraploid and hexaploidy [137].

The leaves of wheat plants stressed with salt alone showed an increase in both Na^+^ and Cl^−^ concentrations; however, the leaves of wheat plants stressed with salt and a high dose of boron showed a significant decrease in Cl^−^ concentration. As a result, under salinity, boron has a positive effect on the concentrations of harmful ions [138,139].

The selectivity of aluminum-activated malate transporter 1 (TaALMT1) for malate over Cl^−^ is complex in wheat varieties resistant to aluminum (Al). Without Al, TaALMT1 is partially active and might be involved in Cl^−^ fluxes across the plasma membrane. The channel becomes less selective for malate than for Cl^−^ at a high external Cl^−^, but the transport is more selective for malate than for Cl^−^ at a low external Cl^−^. The ability of the root tips to excrete malate may be diminished if Al stress and salinity co-occur due to the shift in reversal potential to more negative potentials, depolarization brought on by salinity, and potential competition between Cl^−^ and malate at the cytoplasmic face of the channel [84,140].

### 3.5. Cl^−^ Transporters under Salinity

#### 3.5.1. CLCs Transporting Cl^−^ in Antiport with Protons

A strategy for shoot Cl^−^ exclusion and for regulating cytoplasmic Cl^−^ at a tolerable range involves sequestering of Cl^−^ into the vacuole [84,141,142,143,144,145]. Electrophysiological studies show that the plasma membrane of root-hair cells, beside Cl^−^ channels, contains electrogenic Cl^−^/2H^+^ cotransporters, which mediate either Cl^−^ influx or Cl^−^ efflux across the plasma membrane. There is evidence that the Cl^−^/nH^+^ antiporters also mediate Cl^−^ influx and Cl^−^ effluxes across the tonoplast.

##### CLCs in Maize, Soybean and *Arabidopsis*

It was shown that both maize and *Arabidopsis* confer salt tolerance by regulating Cl homeostasis. *ZmCLC-d* overexpression in *Arabidopsis* improved salt tolerance [146]. A maize CLC gene, *ZmCLCg*, was linked with salt tolerance in a GWAS analysis. Therefore, the salt-hypersensitive phenotype lacking ZmCLCg obtained a prominent Cl^−^ accumulation in shoot tissue [147].

GmCLC1, a vacuolar pH-dependent Cl^−^ transporter in soybean that also functions as a H^+^/Cl^−^ antiporter, controls the accumulation of Cl^−^ in shoots and improves soybean salt tolerance [148,149,150]. The overexpression of wild soybean *GsCLC-c2* enhances Cl^−^ accumulation in the root, enhances Cl^−^ exclusion from the shoot, and raises NaCl resistance [108]. Na^+^ exclusion and salt tolerance are influenced by soybean GmSALT3, an endoplasmic-reticulum transporter in cells connected to the root vasculature. Additionally, it facilitates Cl^−^ exclusion from the shoot and therefore GmSALT3/CHX1 is regarded as a K^+^/H^+^ exchanger [151,152,153].

Mesophyll cells, hydathodes, and the phloem of the leaf exhibit a high expression of AtCLCg, a vacuolar Cl^−^ transporter and homolog of AtCLCc [154].

##### Subcellular CLCs Transporters in *Arabidopsis*

Seven CLCs found in *Arabidopsis thaliana* are localized to the organellar membrane [155]. The spinach AtCLC-f ortholog is localized in the outer membrane of chloroplasts. The subcellular localization and amino acid sequence imply that AtCLCg may be involved in detoxifying the cytoplasm of leaf cells during NaCl stress by sequestering Cl^−^-ions in the vacuole. *GhCLCg-1* overexpression in *Arabidopsis thaliana* improved salt tolerance [110]. In the absence of AtCLCg, a significant amount of Cl^−^ is either in the cytoplasm or in the apoplasm where it inhibits enzyme reactions or increases water loss, respectively [84,156]. When *Arabidopsis* was subjected to NaCl stress, it showed an elevated expression of AtCLCg. This process is crucial because it enables the plant to expel ions when its ion concentration rise too high [157].

The AtCLCg protein regulates the phloem recirculation of Cl^−^, thus preventing the overaccumulation of this ion in photosynthetically active leaves during salt stress. This procedure may help to increase phloem loading and contribute to the translocation of Cl^−^ to petioles, younger leaves, or roots [157].

#### 3.5.2. Nitrate Transporter 1/Peptide Transporter Family (NFP)

The plasma-membrane-localized NPF transports nitrite, glucosinolates, phytohormones, and Cl^−^ and is involved in the influx of 2H^+^:NO_3_^−^ [158]. *AtNPF2.5* encodes a Cl^−^ permeable transporter that regulates root-to-soil Cl^−^ efflux and enhances shoot Cl^−^ exclusion and salt tolerance in *Arabidopsis* [141]. The Zm-NPF6.6 in maize, a NO_3_^−^ preferring transporter, also transports Cl^−^. In maize, the plasma-membrane-localized Zm-NPF6.4 and Zm-NPF6.6 were identified as Cl^−^-preferring transporters [159]. It was demonstrated in *Arabidopsis* that Cl^−^ translocation from root to shoot was accelerated by the rapid downregulation of AtNPF2.4, a Cl^−^ transporter located in the plasma membrane of root stelar cells [142].

##### Accumulation of Cl^−^ in Wheat Is Inhibited by Silicon

The Cl^−^ absorption in the salt-stressed wheat cultivar Vinjett was decreased by silicon supplementation, revealing that Cl transport from roots to shoots is regulated. Additionally, it was noted that the wheat cultivar Vinjett expresses two Cl-transporter genes, TaCLC1 and TaNPF2.4/2.5, in both the roots and the shoots [160]. Since *TaCLC1* was more highly expressed in roots than in shoots, it is most likely that more Cl^−^ is accumulated in the root cell vacuoles and less is transferred to the shoots as a result of Si action. Moreover, the cytosol of leaf mesophyll protoplasts significantly absorbed Cl^−^ under salinity. If Si was supplied before NaCl, this absorption was greatly reduced. This outcome supports the findings of the *TaCLC1* expression analysis, which showed that seedlings treated with Si and NaCl expressed more *TaCLC1* than those treated with NaCl alone. This should improve the sequestration of Cl^−^ into the vacuoles, particularly in shoots. The effects of Si on TaCLC1 and TaNPF2.4/2.5 may be the cause of the decreased concentration of Cl^−^ in the cytosol of leaf mesophyll cells after treatment, although other Cl^−^ transporters might also be involved. It is hypothesized that this makes the wheat plant more resistant to high Cl^−^ concentrations when Si is added. While vacuolar chloride is not harmful, high quantities of chloride present in the cytosol are toxic [160].

### 3.6. Cl^−^ Compartmentalizing Transporters under Salinity

#### 3.6.1. Cation Chloride Cotransporters (CCCs)

The coordinated symport of K^+^, Na^+^, and Cl^−^, is catalyzed by cation Cl^−^ cotransporter proteins. These transporters play a direct or indirect role in controlling root Na^+^ and Cl^−^ absorption in both glycophytes and halophytes.

##### CCCs in Rice, Soybean, Grapevine, and Arabidopsis

In rice, CCCs control the ion homeostasis. K^+^ and Cl^−^ concentrations were lowered when OsCCC1 was knocked out [161]. OsCCC1 in rice is a plasma-membrane-localized transporter and participates in cell elongation.

In *Arabidopsis*, CCC1 is localized to the Golgi and trans-Golgi network (TGN) [162]. Under salt stress, the *atccc* mutants have a higher shoot Cl^−^ [163,164]. VviCCC in grapevine (*Vitis vinifera*) and AtCCC in *Arabidopsis* have been postulated to be involved in long-distance ion transport [162]. Inflorescence stems, roots, leaves, and siliques all had shorter organs when AtCCC was knocked out [42], demonstrating that AtCCC is involved in development and Cl^−^ homeostasis. Grapevine *VviCCC* gene complementation of the *atccc* mutant lowered shoot Cl^−^ and Na^+^ concentrations to wild-type levels under moderate salinity.

## 4. K^+^ Concentrations and Signaling under Salinity

To maintain K^+^ homeostasis, plants need efficient K^+^ absorption pathways, especially under salt stress during which a spike in Na^+^ content is always accompanied by K^+^ loss from roots and shoot tissues, which disrupts the cytosolic Na^+^/K^+^ ratio, a crucial property of plant salt tolerance [165]. The plant cell K^+^ concentration represents K^+^ uptake and efflux. Cell- and tissue-specific K^+^ channels and transporters mediate these activities. This review covers salinity-induced alterations in cellular K^+^ content, uptake, transport, efflux, and signaling cascades.

### 4.1. Cytosolic K^+^ Retention Is Higher in Salt-Tolerant Plants

Plants’ ability to store cytosolic K^+^ during salt stress is crucial to salt tolerance. Smethurst et al. [166] found that salt-tolerant lucerne genotypes retained more K^+^ in their roots. Salt-tolerant types of barley, bread wheat, *brassica*, poplar, and cotton had higher cytosolic K^+^ retention in mesophyll tissues than their salt-sensitive counterparts [167].

### 4.2. K^+^ Channels under Salinity

The identified 77 K^+^-permeable channel genes in *Arabidopsis* were classified into two major groups: 15 genes of K^+^-selective channels (9 Shaker and 6 tandem-pore K^+^ channels) and 62 genes of nonselective cation channels (NSCCs) which include 1 two-pore channel (TPC), 20 cyclic-nucleotide-gated channels (CNGCs), 20 ionotropic glutamate receptors (iGLRs), 10 mechanosensitive-like channels (MSLs), 2 “Mid1-Complementing Activity” channels (MCAs), 1 mechanosensitive Piezo channel, and 8 annexins [168,169,170]. This review examines how salt stress affects the functions of each K^+^ channel.

#### 4.2.1. Shaker K^+^ Channels under Salinity

Shaker K^+^ channels allow enormous, passive K^+^ fluxes through cell membranes. K^+^ channels maintain electrical and osmotic balance. Cellular signaling and metabolic regulation in response to oxidant and salt stressors depend on Shaker K^+^ channels [171]. Shaker channels dominate membrane conductivity in most cells.

##### Shaker K^+^ Channels in Arabidopsis

Nine genes in *Arabidopsis thaliana* encode four functionally distinct plant Shaker K^+^ channels. (i) K_in_ channels: the AKT1, KAT1, KAT2, AKT5, and SPIK (Shaker pollen inward K^+^ channel) genes encode inwardly-rectifying K^+^ influx channels which transport K^+^ into the cell when the plasma membrane is electrically hyperpolarized; (ii) K_out_ channels: the depolarization releases K^+^ from K_out_ channels. GORK (gated outwardly rectifying K^+^ channel) and SKOR (Stelar K^+^ outward rectifier) encode K_out_ channels. (iii) K_weak_ channels: weakly rectifying AKT2/3 channels mediate K_in_-like influx or bidirectional influx/efflux K^+^ channels; and (iv) K_silent_ channels: these channels do not conduct K^+^ currents, but when in hetero tetramers with K_in_ channel subunits, they regulate K^+^ homeostasis. Only KC1 encodes the K_silent_ channels [172,173]. Plant Shaker K^+^ channels appear to play a role in stress-related responses, specifically in osmotic adjustment by maintaining high cytosolic concentrations [174].

##### Shaker K^+^ Channels in Rice and Soybean

It has been demonstrated that the Shaker K^+^ channels OsKAT1 and OsAKT2 in rice promote salt tolerance and protect yield losses from salinity stress [175,176]. Wang et al. [177] reported that *GmAKT1* expression was induced by salinity. *GmAKT1* overexpression in *Arabidopsis* promoted plant growth and enhanced K^+^ concentrations, which decreased Na^+^/K^+^ ratios. *GmSKOR*, *GmsSOS1*, *GmHKT1*, and *GmNHX* revealed increased expression in transgenic *Arabidopsis* expressing *GmAKT1*. Additionally, Feng et al. [178] demonstrated that three K^+^ channel genes (*GmKAT2.1*, *GmSKOR.1*, and *GmGORK.2*) in roots and five genes (*GmAKT6.2*, *GmAKT1.1*, *GmKAT2.1*, *GmSKOR.1*, and *GmGORK.2*) in leaves were elevated in response to the salt stress treatment. *GmAKT1* was markedly elevated in both leaves and roots in response to both salt and drought stress.

At a high concentration of NaCl, ROS increased GORK-mediated K^+^ efflux in rice [179], while NaCl treatment reduced root K^+^ transporters and channel-associated genes (*OsGORK*, *OsAKT1*, *OsHAK1*, and *OsHAK5*). The *OsRbohA* knockout mutant disrupted K^+^ homeostasis. Overexpressing *OsRbohA* boosted K^+^ transporter and channel gene expression, reducing root K^+^ loss [180].

#### 4.2.2. Tandem-Pore K^+^ Channels

During stress, three K^+^ channels: slowly activating channel (SV), fast vacuolar channels, and vacuolar K^+^ channel (VK) transport K^+^ between the cytoplasm and vacuole [181]. Maathuis et al. [182] found higher SV-channel activity in leaf vacuoles from the extreme halophyte *Suaeda maritima* cultivated in highly salinized circumstances. K^+^ transport blockage may cause osmoregulation issues in salt-stressed plant cells [26]. A rise in luminal Na^+^/K^+^ ratio reflected Na^+^ accumulation in vacuoles under salt stress and changed the threshold for SV activation to positive potentials, reducing the SV channel open probability under saline circumstances. Under salt stress, plants regulate SV channel activity differently to limit Na^+^ leakage into the cytoplasm [183]. Choi et al. [184] were the first to show that TPC1 affects the velocity of the salt-induced Ca^2+^ wave in *Arabidopsis* and salt-stress-induced Ca^2+^ waves in the *tpc1-2* knockout mutant, although they were weaker, whereas *TPC1* overexpression accelerated the wave.

#### 4.2.3. Nonselective Cation Channels (NSCCs) under Salinity

The fundamental entry points for K^+^ extrusion and sodium ion uptake in roots are NSCCs [185]. Plant NSCCs are widespread at the plasma and tonoplast membranes.

With regard to K^+^ efflux in response to salt stress, depolarization-activated nonselective activation channels (DA-NSCCs) play a significant role [46]. Wu et al. [186] demonstrated a salt-induced K^+^ efflux from wheat mesophyll cells similar to that in *Arabidopsis.* When roots are exposed to salinity, the strong inwardly directed electrochemical Na^+^ gradient drives a major flux of Na^+^ into root cells, possibly via NSCCs, and this results in a significant plasma-membrane depolarization [187,188,189]. The depolarization-activated GORK channels react to this alteration quickly and facilitate the enhanced K^+^ efflux. The ROS-induced NSCC for K^+^ exclusion in pea roots was demonstrated by Bose et al. [190]. ROS, created when cytosolic K^+^ decreases and cytosolic Ca^2+^ increases, further activate ROS-producing NADPH oxidases and this amplifies GORK-mediated K^+^ efflux under high salinity stress. This is the molecular mechanism enabling the K^+^ efflux mediated by ROS-activated NSCCs [191].

#### 4.2.4. Cyclic Nucleotide-Gated Channels (CNGCs) and Glutamate-Like Receptors (GLRs)

The CNGCs and GLRs are two major NSCCs in plants. Under salinity, at least some CNGCs exhibit equal permeability for K^+^ and Na^+^, which may have an effect on cytosolic K^+^/Na^+^ ratios [192]. Mutants such as *atcngc3* and *atcngc10* of *Arabidopsis* and *oscngc1* of rice showed better growth by preventing the excessive inflow of Na^+^ [193,194,195].

Plant GLRs function as NSCCs, much like the ionotropic glutamate receptors (iGluRs) in mammalian cells and are ligand-gated channels permeable to Ca^2+^, K^+^, and Na^+^. Under salt stress, the glutamic acid gradient in the root tips is enhanced to a high level which functions as an activator of GLRs [46]. When exposed to salt stress, glutamate generated in the root tips stimulates GLRs, which then activate the plasma membrane’s NADPH oxidase to produce too much hydrogen peroxide (H_2_O_2_). This further activates outward-rectified K^+^ channels and results in programmed cell death [196].

Transgenic plants which overexpress *AtGLR2* are more sensitive to K^+^ and Na^+^ salt. All 20 *AtGLR* genes have significant levels of expression in *Arabidopsis* roots, suggesting that AtGLRs are crucial for controlling ion intake from the soil, especially K^+^ uptake [197]. The Ca^2+^-dependent protein kinase (CDPK) phosphorylates AtGLR3.7 and influences Ca^2+^ signaling, and the *atglr3.7-2* mutant is hypersensitive to salinity [198]. AtGLR3.4 mediates Na^+^ accumulation in germinating seeds under salt stress because the ratio of Na^+^ to K^+^ is higher in *atgl3.4-1* seedlings than in wild-type seeds under salt stress [199].

#### 4.2.5. Two-Pore K^+^ Channels under Salinity

Vacuolar TPK channels are both mechanosensitive and osmo-sensitive. They may serve as cellular osmo-sensors during abiotic stresses [200]. Under salt stress, VK channels provide two beneficial functions: they export K^+^ from the vacuole to increase the cytosolic K^+^/Na^+^ ratio and act as a shunt conductance for H^+^-pumping [201]. *Arabidopsis* TPK1, maintains K^+^ homoeostasis under salinity stress by closing stomata via KIN7 kinase and assisting plants in resisting salt tolerance [202]. Significant salt-induced variations in the transcripts of *TPK (VK*) were observed [203].

When exposed to salt stress, Ca^2+^-dependent protein kinases (CDPKs) phosphorylate the TPK1 vacuolar K^+^ channel, and both the *cpk3* and *tpk1* mutants showed salt-sensitive phenotypes. *TPK1* and *CPK3* are constitutively co-expressed, and Ca^2+^ activates them both. Salt stress greatly induces VK/TPK1, and interactions between the cytoplasmic N-terminal region and 14-3-3 proteins increase the probability of VK channel opening [204]. Since NtTPK1 shows considerable selectivity for K^+^ over Na^+^, the fact that tobacco *TPK1* expression increased under salinity suggests that this protein may be involved in delivering K^+^ into the cytosol [203].

##### TPKs in Rice

Rice OsTPKa is significantly increased under salinity, but OsTPKb shows no noticeable change [205]. Ahmad et al. [206] found that the overexpression of *OsTPKb* improves the K^+^ ratio between the cytosol and small vacuole which confers the ability to withstand osmotic stress.

### 4.3. K^+^ Transporters under Salinity

The high-affinity K^+^ (HAK)/K^+^ uptake permeases (KUP)/K^+^ transporter (KT) family, K^+^ and/or Na^+^ selective high-affinity K^+^ transporter (HKT), and cation: proton antiporter (CPA) family are the three major K^+^ transporter families in plants [196,207].

#### 4.3.1. HAK Transport Systems under Salinity

During the early stages of salinity, the HAK/KUP/KT transport systems play a major role in K^+^ absorption [167]. Numerous plants such as corn [208], barley [209], rice [210], tomato [211], peach [212], pear trees [213], *Ipomoea* [214], tea [215], cotton [180], and *Arabidopsis* [201] have been shown to contain HAK/KUP/KT transporters. In addition to their crucial roles in K^+^ absorption and transport, the HAK/KUP/KT family is shown to confer tolerance to both salt and drought stresses.

##### HAKs in Rice, Maize, *Medicago* and Pepper

When rice *OsHAK5* was expressed in tobacco BY2 cells, it was shown that the cells accumulated a significant quantity of K^+^ rather than Na^+^ when exposed to salt stress. This finding suggests that OsHAK5 is a salt-sensitive high-affinity K^+^ transporter that imparts enhanced salt tolerance [216]. The K^+^ to Na^+^ ratio and salt tolerance are decreased by rice *OsHAK21* knockout mutation [217]. *ZmHAK4* is selectively expressed in the root stele in maize and is involved in removing Na^+^ from xylem sap. Furthermore, ZmHAK4 exerts specific roles in increasing shoot Na^+^ exclusion and salt tolerance when used in combination with ZmHKT1 [218]. Song et al. [219] in rice showed that an ER-localized OsCYB5-2, a cytochrome protein can bind to OsHAK21, enhances OsHAK21-mediated K^+^ absorption and improves salt tolerance. *MtHK2/7/12* from *Medicago truncatula* and *MsHAK2/6/7* from *Medicago sativa* were significantly induced by salt stress [220]. Pepper *CaHAK3* and *CaHAK7* transcript abundance was strongly and specifically upregulated in pepper roots under low K^+^ or high-salinity conditions, suggesting the crucial roles for conserving the Na^+^/K^+^ balance during salt stress in pepper [221].

#### 4.3.2. KUP-Transport Systems under Salinity

In several plants, varying number of KUPs were identified [222]. Ions, notably Cl^−^ and Na^+^, that accumulate excessively hamper certain transport systems, including the KUP transporters [223]. The abundance of *McKUP1* and *KUP4* homologous transcripts increased in common ice plants with salt exposure and K^+^ deficiency [224]. *KUP2* transcription is downregulated in the shoots of salt-stressed plants [225]. During salt stress, the expression levels of both *KUP6* and *KUP11* transcripts in root tissue are upregulated [226]. The KUP7 transporters [227] or depolarization-activated K^+^ channels or transporters like SKOR [228] should mediate the xylem K^+^ loading when hyperpolarization-activated AKT channels are suppressed during the onset of salinity.

#### 4.3.3. HKT-Transport Systems under Salinity

Both K^+^ starvation and NaCl stress regulated plant HKTs [226]. Many plant HKT transporters mediate the influx of Na^+^, and some are Na^+^/K^+^ symporters [229].

##### HKTs in Wheat, Rice, Barley, Soybean, and Tomato

Wheat TaHKT2;1 mediates the efflux of Na^+^ in root tissue like rice OsHKT2;1 [230]. Salt and K^+^ deprivation trigger TaHKT2;1 [229]. Furthermore, HKTs such as TmHKT1;5-A in einkorn wheat [228], HvHKT1;5 in barley [231], SlHKT1;1 and SlHKT1;2 in tomato [232], GmHKT1;4 in soybean [233], and TaHKT1:5-D in bread wheat [234] function diversely and improve plant K^+^ use efficiency.

An enhanced action of OsHKT1;5 leading to Na^+^ unloading from the xylem stream was observed [37]. Suppressing rice *OsHKT2;1* gene may make plants tolerate salt [235], and salt sensitivity results from root-specific OsHKT1;5 deletion [236]. OsHKT2;4 is likely a K^+^ transporter rather than a Na^+^/K^+^ symporter and helps to maintain a greater K^+^ level even in salinity [237]. The transcriptional proteins OsSUVH7, OsMYB106, and OsBag4 work together to upregulate OsHKT1;5 in roots during salt stress [238]. Disrupting OsSUVH7 or OsMYB106 binding sites in the *OsHKT1;5* promoter or knocking down these genes affects the expression of *OsHKT1;5* in roots and ultimately rice salt tolerance [236]. After salt stress, *osbhlh044* CRISPR/Cas9 mutants had more Na^+^ and less K^+^, resulting in a higher Na^+^/K^+^ ratio [239].

The salinity-induced higher expression of barley *HvHKT1;5* and *HvHKT2;1* [240,241] and maize *ZmHKT2* [242] results in enhanced plant growth and K^+^ concentration, lowering salt stress by lowering Na^+^/K^+^ ratios, and ultimately leading to salt tolerance. Transgenic *Arabidopsis* expressing soybean K^+^ transporters such as GmHKT1, GmSKOR, GmsSOS1, and GmNHX1 and rice OsHKT2;1 also exhibited higher salt tolerance by higher K^+^/Na^+^ ratios.

### 4.4. Exchangers and Antiporters under Salinity

Plant monovalent cation-proton antiporters (CPAs) are shown to regulate K^+^ homeostasis and are classified into two types: NHAP/SOS and NHE/NHX transporters (CPA1) and K^+^ efflux antiporters, KEAs (CPA2). CHXs—cation/H^+^-exchangers (CHX)—are also important [45,63,243].

#### 4.4.1. CPA1 Transporters in Transgenic Arabidopsis, Sugar Beet, Rice, and Cotton with Increased Salt Tolerance

The H^+^-linked K^+^ transport at the tonoplast and the K^+^ distribution between the vacuole and cytosol are processes that are mediated by NHX proteins [244]. *AtNHX1*-carrying transgenic plants clearly displayed salt tolerance [245]. Transgenic *Arabidopsis* expressing wheat *NHX1* and date palm *PdNHX6* displayed a balanced Na^+^/K^+^ ratio and transgenic sugar beet expressing *AtNHX3* has been demonstrated to promote salt tolerance [246]. Additionally, increased salt tolerance has been observed by the transgenic expression of the *NHX1* gene from one genus in another [247]. When *AtNHX1* is overexpressed, plasma-membrane-localized *GhSOS1* expression levels in cotton roots are considerably elevated [248]. It was discovered that *NHXs* from different plants have varying expression patterns in various plant organs due to salt stress [249]. The expression of *OsNHX1*, *OsNHX2*, *OsNHX3*, and *OsNHX5* in rice triggered by salt stress in different tissues plays a significant role in conferring salinity tolerance [250].

##### K^+^ Efflux Antiporters, KEAs (CPA2) Mediate Both K^+^ Influx and Efflux

*Arabidopsis* K^+^ efflux antiporters (KEAs) are comparable to *Escherichia coli* KefB and KefC. The chloroplasts of a *kea1 kea2* double mutant, as well as a *kea1 kea2 kea3* triple mutant, are swollen and shorter, which make mutants susceptible to salt stress [251]. *Escherichia coli* cells expressing *KEA1*, *KEA2*, and *KEA3* are hypersensitive to even mild salt stress. It is suggested that KEA4-KEA6 may improve salinity tolerance by regulating the pH of the endomembrane network [252]. Overexpression of *KEA4*, *KEA5*, or *KEA6* in the *kea4 kea5 kea6* mutant abolished salt susceptibility. Under salinity, the *kea4 kea5 kea6* expressing *NHX1* or *NHX2* grew better than the triple mutant. Coordinated actions of *KEAs* and *NHXx* in specific combinations result in enhanced salt tolerance [67].

### 4.5. CHX Transporters in Arabidopsis

Plant CHX transporters mediate the transport of K^+^, Na^+^, H^+^, and Cl^−^ [45]. CHX transporters are essential for cytosolic K^+^ level maintenance and producing K^+^ signatures during salinity stress [191]. In the cortical cells of adult root zones and epidermal cells, *Arabidopsis AtCHX17* was expressed preferentially and was induced by salt stress, indicating that AtCHX17 may also be involved in K^+^ uptake and homeostasis during salinity. *AtCHX17* knockout mutants accumulated less K^+^ in roots than the wild type [253]. *Arabidopsis* transgenic lines overexpressing soybean *GsCHX19.3* displayed a reduced Na^+^ concentration and higher K^+^/Na^+^ ratios during salt and alkaline conditions and produced plant resistance to both stressors [254].

### 4.6. K^+^ Signaling under Salinity

In addition to being an essential nutrient, K^+^ has multiple roles, namely ion signaling, enzyme catalysis, water osmoregulation, and stress tolerance [255]. To establish salt resistance mechanisms, plants are compelled to consume the maximum fraction of the available ATP [256]. Salinity-induced transient cytosolic K^+^ efflux activates a metabolic modification which limits energy-draining anabolic reactions and conserves energy for a strategic advantage under the salinity-induced energy-limiting conditions [171]. Importantly, K^+^ efflux caused by salt does not compete with the nutrient requirement of plants for K^+^. The transient nature and substantial tissue specificity of stress-induced K^+^ efflux finely regulates this [46].

Various plants adopt various patterns of K^+^ signaling that exhibit commonalities with cytosolic Ca^2+^ signaling [35]. Transient Ca^2+^ surges in pants under salinity are processed by a variety of protein kinases and Ca^2+^-binding proteins [257]. Activation of the Ca^2+^ signaling cascade, which includes the CBL-CIPK complex and CDPKs, is one of the key regulatory mechanisms for K^+^ uptake and transport in plants [258]. The salt tolerance of transgenic rice was improved by the overexpression of bermudagrass *CdtCBL4* (calcineurin B-like protein 5) and *CdtCIPK5* (CBL-interacting protein kinase 5). Under salt stress, Na^+^/K^+^ homeostasis was regulated by CBL1 in conjunction with CIPK24, CIPK25, and CIPK26 in poplar [259]. Hence, the CBL-CIPK signaling network regulates K^+^ signaling to increase plant salt tolerance.

An et al. [260] further showed that alfalfa MsCBL4 not only amplified the activities of antioxidant enzymes during salt stress, but also accumulated Ca^2+^ and decreased the Na^+^/K^+^ ratio in roots, as well as the level of ROS, to overcome the oxidative damage. As a result, crosstalk between Ca^2+^, K^+^, and ROS could improve plant salt resistance. By enhancing stomatal conductance and raising the CO_2_ content in leaf tissue, the accumulation of K^+^ may restrict the production of ROS during salt stress, lowering lipid peroxidation and promoting tomato growth [261,262].

The transcription and post-translation of plasma membrane H^+^ ATPase to increase K^+^ uptake in pumpkin root tips was mediated by RBOH-dependent H_2_O_2_ signaling, which improved salt tolerance [39]. The mangrove tree, *Kandelia obovate*, is better able to adapt to high salinity because nitric oxide (NO) increases its capacity for K^+^ uptake, lessens the cytosolic K^+^ loss, and activates AKT1 K^+^ channel and NHX antiporters to maintain K^+^/Na^+^ equilibrium [263].

Under salinity, a lower Na^+^/K^+^ ratio and increased ROS production in the roots of mung bean, an increase in net K^+^ and Ca^2+^ fluxes caused by H_2_O_2_ in the roots of barley, a higher K^+^/Na^+^ ratio in the leaves of tomato, a decrease in Na^+^/K^+^ homeostasis and root K^+^ loss in the leaves and roots of brome grass overexpressing *BdCIPK31*, and ROS-induced K^+^ efflux and Ca^2+^ absorption in barley reflect the cross talk among K^+^, Ca^2+^, and ROS signaling cascades which collectively enhance plant salt tolerance [264].

## 5. Cytosolic Ca^2+^ Signaling

Ca^2+^ is an important second messenger under both abiotic and biotic stresses, and is also involved in cell division, polarity, growth, and development [265]. Salinity causes both osmotic stress and ionic toxicity in plants, which induce various Ca^2+^ “signatures”, free cytosolic concentration changes, leading to different downstream reactions. Salt-stress-activated Ca^2+^ elevation is connected with an increase in reactive oxygen species, ROS, and ABA concentrations [266,267].

### 5.1. Ca^2+^ Transport System

The influx and efflux of Ca^2+^ are mediated by a complex system of proteins, such as channels, antiporters, and ATPase-mediated pumps [268,269]. Transport in a plant cell is shown in Figure 4.

The increase in cytosolic free Ca^2+^ concentration, [Ca^2+^]_cyt_, depends on Ca^2+^ transport from the apoplast or from internal stores, such as the vacuole, ER, or mitochondria [270,271]. Under salinity, plants transport Ca^2+^ into the cytosol by channels or protein transporters; in *Arabidopsis*, cyclic-nucleotide-gated channels, AtCNGCs, and glutamate-receptor-like receptors, AtGLRs, mediate Ca^2+^ influx [272]. Ca^2+^ changes under salt stress can also be due to transport via two-pore channels, TPCs, in the tonoplast, reduced hyperosmolarity-induced [Ca^2+^]_cyt_ increase channels, OSCAs, which transport calcium under osmotic stress, and three types of mechanosensitive-like channels, MSLs [273,274]. Calcium is transported out from the cytosol mainly by pumps or Ca^2+^/H^+^ antiporters in an active and secondary active way, respectively [268,269].

#### 5.1.1. Ca^2+^ Transport by Channels

Ca^2+^ is taken up into the cytosol by many different channels: HACCs, hyperpolarization-activated cation channels, DACCs, depolarization-activated cation channels, and other NSCCs, nonselective cation channels, such as CNGCs, cyclic-nucleotide-gated channels, and GLRs, glutamate-receptor-like channels, and also MLRs, mechanosensitive-like channels.

Several Ca^2+^ channels also are located in the endomembranes [268]. Ca^2+^ is transported out from the vacuole by three channels: SV, slow-activating vacuolar channel; RyR, cyclic ADP-ribose-activator ryanodine receptor-like channel; and InsP_3_R, inositol 1,4,5-trisphosphate receptor-like channel. Ca^2+^ is transported out from the ER by the channels NAADP, InsP3R, and RyR, but by FACCs, fast-activating cation channels, into the chloroplast. In the nucleus, Ca^2+^ is transported by NSCCs. Within the mitochondria, Ca^2+^ is moved by an antiporter and a uniporter.

Most investigated of the Ca^2+^-influx channels are the CNGC channels (Figure 5). They are regulated by the binding of cGMP and cAMP and phosphorylation [269]. Reports show that CNGCs are selective for Ca^2+^ uptake and are located mainly in the PM and tonoplast of the root epidermis and leaf mesophyll cells. Experiments showed that some of them are activated when co-expressed with protein kinase CPKs.

#### 5.1.2. Ca^2+^ Transport from the Cytosol and Chloroplast and into ER, Golgi, and Vacuole

Ca^2+^ is actively transported from the cytosol by the Ca^2+^ATPase ACA and by secondary active Ca^2+^/H^+^ antiporters, CAXs. ACAs also transport Ca^2+^ into the ER, vacuole, and Golgi, but out from the chloroplast. ECAs, ER-type Ca^2+^ATPases, are only localized in the endomembranes, such as the ER. They pump Ca^2+^ into the ER. The Ca^2+^ATPases are high-affinity Ca^2+^ transporters that shape the Ca^2+^ concentration to a resting value (50–100 nM) after a Ca^2+^ elevation. The Ca^2+^/H^+^ antiporters are low-affinity Ca^2+^ transporters that transport higher concentrations of Ca^2+^.

In the tonoplast, Ca^2+^/H^+^ antiporters, the Ca^2+^ exchangers, CAXs 1–6, with a lower affinity for Ca^2+^ than the Ca^2+^ATPase, mediate Ca^2+^ transport [275]. CAX1 and CAX3 are important for keeping Ca^2+^ homeostasis. It was reported that in *Arabidopsis*, CAX1 is more highly expressed in shoot and *CAX3* more highly expressed in roots. It is proposed that the co-expression of the *CAX1-CAX3* complex is involved in salt tolerance [276].

### 5.2. Ca^2+^ Signals Depend on the Type of Stress, Transporter Location and Type, and Duration of Stress

Measurements of free [Ca^2+^_cyt_] in single protoplasts of rice, wheat, and beans by the use of dual-wavelength fluorescence microscopy and the Ca^2+^ indicator Fura2, AM [277,278,279], show that the rise in Ca^2+^ is usually lower than in intact plants, such as transgenic *Arabidopsis* [280,281]. This may depend on the lack of cell walls in protoplasts, as cell walls can also be involved in Ca^2+^ signaling [281]. The extracellular domain of the protein FERONIA is connected with cell-wall pectin and supposed to take part in Ca^2+^ signaling under salt stress [178]. The excess of ions in the cell walls might cause structure changes that can be sensed by glycoproteins and plasma-membrane-localized receptor kinases and cell-wall-associated kinases, WAKs. WAKs can also bind both pectin and Ca^2+^, and an excess of Na^+^ can affect the binding and induce Ca^2+^ signaling [24]. In addition to the cytosol and apoplast, cell organelles and the endomembrane system might also participate in Ca^2+^ signaling. In some of the organelles, Ca^2+^ fluxes are connected with the cytosol, while in others the signaling is independent of the cytosol. We will not discuss organellar signaling in this review as there is still scarce information on this topic.

#### Dynamics of the Cytosolic Ca^2+^ Signals

Ca^2+^ signals under salinity are usually transient, or sustained, and in the latter case Ca^2+^_cyt_ can be oscillating or remain for a long time [64,271,279]. Other types of abiotic stress show different Ca^2+^ signals, and different channels are involved in the Ca^2+^_cyt_ increase [49]. Oxygen-deficiency stress induces Ca^2+^_cyt_ elevation during a long time, 10 min or more [282]. Under chilling stress in tomato, quite different [Ca^2+^_cyt_] signals were monitored depending on the cells’ resting level of Ca^2+^ under non-stress conditions [283]. In *Arabidopsis*, salt stress in roots induced transient Ca^2+^ waves through the whole plant with a speed of 400 µm/s [184]. The dynamics of the Ca^2+^_cyt_ signaling depend on both influx and efflux mechanisms of Ca^2+^ and stress duration.

The Ca^2+^ signal often depends on the external Ca^2+^ concentration. In the presence of 0.1 mM external Ca^2+^, rice cultivars with different salt tolerances demonstrated a higher increase in [Ca^2+^_cyt_] in mesophyll protoplasts of the salt-tolerant cv. Pokkali than in mesophyll protoplasts of sensitive cv. BRRI Dhan29 [277]. Salt addition in the presence of 1.0. mM Ca^2+^ did not change [Ca^2+^_cyt_] in the tolerant cultivar but increased it in the sensitive one. Since 1.0 mM Ca^2+^ is known to block NSCCs, it was concluded that this type of channel was present in cv. Pokkali. Pharmacological experiments showed that the apoplast was the main store for Ca^2+^ elevation in the sensitive cultivar, but in the tolerant cultivar, Ca^2+^ was mainly transported from internal stores. Contrary to ionic stress, which causes an increase in [Ca^2+^_cyt_], osmotic stress in rice caused a decrease.

### 5.3. Possible Na^+^ Sensors

#### 5.3.1. The Na^+^/H^+^ Antiporter SOS1 Is Critical for Cytosolic Ca^2+^ Elevation, as Shown in *Arabidopsis*, Soybean, and Rice

Few investigations have dealt with Na^+^_cyt_ influx into *Arabidopsis* mesophyll cells. A recent study compared cytosolic Na^+^ influx in *Arabidopsis* Wt, sos1;1, and nhx1 mutants and the induced cytosolic Ca^2+^ changes [64]. By epi-fluorescence microscopy and fluorescent probes, specifically binding to Na^+^ and Ca^2+^, SBFI, AM, and Fura 2, AM (Figure 2), a higher Na^+^_cyt_ influx was demonstrated in both mutants compared with Wt when 100 mM NaCl was added. In the Wt, the influx was low and mainly transient, but in the other genotypes the influx continued for a longer time showing different dynamics. The addition of NaCl induced Ca^2+^_cyt_ elevation in the Wt, but not in the sos1;1. A small increase in Ca^2+^ was also obtained in nhx1. Pretreatment of the protoplasts with LiCl before NaCl addition caused a strong inhibition of Ca^2+^ elevation. The results suggest that the salt-induced Ca^2+^ elevation depends on influx from both internal and external stores and is obtained in the presence of an intact Na^+^/H^+^ antiporter.

In soybean, *Glycine max*, the GmSOS1 in roots is critical for salt-stress tolerance [221]. The mutant gmsos1, constructed by the CRISPR-Cas9 system, accumulated more Na^+^ in the roots of the soybean than the Wt and increased K^+^ efflux under salt stress. Moreover, the SOS1 in rice was also considered important for salt tolerance [284].

#### 5.3.2. GIPC Can Bind Na^+^

It is still uncertain how plants sense Na^+^. Recently, it was reported that the monocation-induced [Ca^2+^] increases 1, MOCA1, encoding a glucuronosyl transferase, involved in the biosynthesis of a sphingolipid, glycosyl inositol phosphorylceramide, GIPC, was required for Ca^2+^ signaling under ionic stress. The findings suggest that Na^+^ can bind to the plasma-membrane-located GIPC, and that Na^+^ is transported into the cytosol and activates an unknown Ca^2+^ channel [11,285] (Figure 6).

#### 5.3.3. A Proposed Structure and Function of SOS1

SOS1 is a protein with 12 transmembrane domains and is the longest ion transporter in the plant plasma membrane with 1146 amino acids [286]. It is proposed to be important for salt stress tolerance as it can transport Na^+^ out of the cytosol by its Na^+^/H^+^ antiporter. It was recently demonstrated that Na ions can bind to GIPC, a glycosyl inositol phosphoryl ceramide sphingolipid in the plasma membrane [11]. The model is described by Xie et al. [286] suggesting that Ca^2+^ can be transported by the same lipids. In our model (Figure 6), we suggest that Na^+^ is both bound to the GIPC and transported into the cytosol by this sphingolipid, and that Ca^2+^ instead is transported via a Ca^2+^ channel close to GIPC, maybe induced by a membrane potential change or conformational changes in the protein. One reason for this is that our previous results from rice indicated that Na^+^ is transported into the cytosol before a Ca^2+^ elevation is induced [287]. In *Arabidopsis*, the annexin channel AtANN4 was shown to transfer Ca^2+^ under salt stress, but other channels may also be involved. Some reports show that annexins are cytosolic proteins that are associated with PM phospholipids [288,289]. The CNGCs, cyclic nucleotide-gated channels, were reported to induce selective Ca^2+^ transport under salt stress and might also be a candidate channel in salt-stress signaling [290].

Ca^2+^ is sensed by two proteins with a similar function in the cytosol, the CBL10/SOS3, calcineurin B-like protein 10, and SCABP8, Ca^2+^-binding protein 8, activated by Ca^2+^ binding and phosphorylation by binding to the protein kinase SOS2, SOS2/CIPK24, or CIPK8 [286]. The SOS3/SOS2 and SCABP8/SOS2 can transfer SOS2 to the 1136–1138 inhibitory domain of SOS1 protein and release the inhibition of the SOS1. Na^+^ can then be transported to the apoplast by the Na^+^/H^+^ antiport. For easier reading, only SOS3, SCABP8, and SOS2 are shown in Figure 6.

SOS2 can also activate the K, Na^+^/H^+^ antiporter NHX in the tonoplast, and CAX, Ca^2+^/H^+^ antiporter in the PM [291].

#### 5.3.4. ANN1, KEAs and FERONIA and Other Ca^2+^ Transporters

Other suggested sensors of high salt that might cause Ca^2+^ signaling are the plasma membrane localized annexin1, ANN1, plastid-K exchange antiporters, KEAs, and FERONIA (FER) proteins [251,288,292]. At high external Na^+^, annexin1 mediates a ROS-activated Ca^2+^ influx into the cytosol. A mutation in the ANN1 gene causes an increase in Na^+^ and diminishes plant growth. KEAs might function as sensors of osmotic stress, but mechanosensitive-like proteins, MSLs, can also react to changes in the turgor pressure, which is connected with osmotic stress. FER belongs to a receptor-like kinase-like (RLKiL) protein family that binds to pectin. A loss of function of this protein results in the inhibition of salt-triggered Ca^2+^ signaling. In the plasma membrane of root cells, mechanosensitive-like channels, MSLs, but also protein kinases, Ca^2+^ channels, e.g., NSCCs, cyclic nucleotide-gated channels, CNGCs, and receptor-like kinases, RLKs, have been suggested to be sensors of salinity.

## 6. Ca^2+^ Signal Transmission into Intracellular Downstream Reactions

Not only Ca^2+^ but also NO, ROS, phosphatidic acid, and cyclic nucleotides might serve as secondary messengers in stress reactions and make them very complex [292]. Ca^2+^ signaling is often connected with pH changes, but also with K^+^ signaling [293]. Plants respond to stress by activating various signaling pathways that allow them to escape from the stress or acclimate to it. The specific stress-induced Ca^2+^ elevation is recognized by different Ca^2+^-binding proteins, which transmit the signal into a cascade of reactions that involve protein phosphorylation, protein translocation, and gene expression [257,294].

The Ca^2+^ signals are decoded by many types of sensors, such as calmodulin, CaM, CaM-like proteins, CMLs, calcineurin B-like proteins, CBLs, and Ca^2+^-dependent protein kinases, CIPKs [266]. Ca^2+^ can bind to a domain on the sensor called an EF-hand motif: elongating factor-hand motif. Sensor proteins can function as both sensor relays and sensor responders. The major calcium sensors are the calcineurin B-like proteins, CBLs. These proteins interact with protein-kinases, CIPKs, to form a complex CBL-CIPK network in order to combat the specific type of stress by different mechanisms (Figure 7). Stimuli can rapidly produce an increase in cytosolic Ca^2+^, [Ca^2+^]_cyt_, but also the production of ABA, abscisic acid, and ROS, reactive oxygen species. An increase in Ca^2+]^_cyt_ might also occur from ROS and ABA formation. ABA is necessary for growth and development, and also for surviving salt, drought, and cold stresses. Under ABA-dependent stress, Ca^2+^ is suggested to be sensed by both CBL1 and CBL9. These proteins bind to CIPK15 and CIPK3, respectively, which activate ABI1/2, ABA-insensitive 1/2, and ABR1, abscisic acid repressor 1, which are involved in ABA signaling.

The CBL-CIPK pathway is the major signaling pathway under abiotic stress, such as salinity stress, leading to Na^+^, H^+^, and K^+^ homeostasis, and ROS signaling [266]. A simplified model of important pathways under salinity is shown in Figure 7.

## 7. pH Changes and Signaling under Salinity

It has been well known since the nineties that stress in higher plants induces changes in pH_cyt_ that often occur together with [Ca^2+^_cyt_] changes [295,296]. The pH changes in the cytosol of cells or protoplasts can be measured by dual-wavelength fluorescence microscopy and the pH-sensitive dye BCECF, AM, an acetoxymethyl ester of carboxyfluorescein, while vacuolar pH can be monitored with confocal microscopy and 6-carboxyfluorescein or Origon Green dextran dyes [277,279].

### 7.1. PH Signaling under Salinity

Okazaki and coworkers [297] demonstrated that NaCl stress triggers fast changes in [Ca^2+^_cyt_], pH_cyt_, and vacuolar pH, pH_vac_, in a charophyte alga. The pH changes in rice were connected with Ca^2+^_cyt_ signaling in a complex way depending on the concentration of the stressor, duration of the stress, and sensitivity of the species, organ, or cell type [57]. In some cases, the pH changes also occur independently of Ca^2+^ changes. It is expected that protons have a role in signaling as there are steep concentration differences both of Ca^2+^ and protons within the plant cell. Under normal conditions, the pH is around 7.5 in the cytosol, but in the vacuole lumen and apoplast, the pH is much lower, around 5.5 [298]. pH regulation in a cell is dependent on the proton pump and antiporter activities, but other mechanisms are also needed. Recent findings suggest that CCC1, a cation-chloride cotransporter in *Arabidopsis*, is necessary for the regulation of pH in the Golgi-network early endosome [299]. If this regulation does not function, the plant cannot develop normally, it will obtain reduced size, chlorophyll defects, and stem necrosis and cannot respond to abiotic stress [42].

### 7.2. Salinity Induces Different Cytosolic and Vacuolar pH Changes in Salt-Sensitive and Salt-Tolerant Species or Cultivars

In the salt-tolerant halophytes quince and quinoa and in tolerant rice, salt stress caused an increase in cytosolic pH but a decrease in vacuolar pH [52,57,58,277]. In the halophyte quinoa, pH_cyt_ increased from 7.5 to 8.5 upon salt addition and was then stable at 7.7. No reaction occurred upon the addition of salt to protoplasts from salinity-cultivated quinoa, but these protoplasts had a somewhat higher resting pH_cyt_ than the ones cultivated without salinity. In the root epidermis of quinoa, salinity treatment activated the PM H^+^-ATPase [76].

Since the increase in pH_cyt_ in tolerant rice was inhibited by NH_4_NO_3_, an inhibitor of V^+^ATPase in the tonoplast, it was suggested that protons were transported from the cytosol into the vacuole in tolerant plants. This was also confirmed by expression analysis of VHA, which showed an increased expression in rice after salt-stress cultivation [33].

In sensitive plants, like pea, field beans, and sensitive rice, the opposite reactions occurred, pH_cyt_ decreased and pH_vac_ increased after salt treatments [58,82,277]. In this case, the pH_cyt_ decrease was inhibited by NH_4_ VO_3_, an inhibitor of H^+^ATPase at the plasma membrane, and could be caused by activation of the Na^+^/H^+^ antiporter SOS1 in the plasma membrane, which is energized by the PM H^+^ATPase [300,301]. Experiments with leaves from salt-sensitive tobacco and potato plants that were transformed by a gene for a pH-sensitive protein, Pt-GFP, also reacted in the same way: salt treatment caused a decrease in pH_cyt_, and depended on the time the plants were subjected to the stressor [302].

The different reaction in quinoa when salt was added might depend on some kind of acclimation, probably by activation of the H^+^ATPase, which was absent in pea [56].

Transient pH changes likely occur by Na^+^/H^+^ antiporter or H^+^/Cl^−^ symporter activities. The salinity-induced acidification in pea might result from the salt-stress inhibition of the H^+^-ATPase activity by Ca^2+^-activated protein kinases [303], but also from Cl^−^/H^+^ co-transport.

Reports show that halophytes can maintain a high negative membrane-potential difference under salinity, necessary for keeping a stable cell [K^+^_cyt_] and to exclude Na^+^. The work of [69] demonstrated that the high salt tolerance in halophytes, compared with glycophytes, does not depend on a higher AHA transcript level but their ability to rapidly activate the PM H^+^-ATPase upon salt treatment.

### 7.3. Extra Addition of Calcium or Potassium under Cultivation of Wheat and Field Beans in Saline Medium Affects the Cellular pH and H^+^-ATPase Activity

In addition to being an important second messenger under stress, calcium is necessary for the normal growth and development of plants [79]. Potassium is an important nutrient too, since this element is a co-factor for many enzymes and used for osmotic regulation.

Salinity can cause both Ca^2+^ and K^+^ deficiencies in plants under cultivation. When the soil K^+^ concentration is low, a high Na^+^ level will compete with K^+^ for the same uptake sites as these ions have a similar chemical structure. One strategy is to add extra CaSO_4_ and/or K_2_SO_4_ to the root media.

An essential premise for plant growth is a low apoplastic pH and this depends on an active H^+^-ATPase that pumps protons from the cytosol into the apoplast [304,305]. A high external pH triggers a cytosolic Ca^2+^ increase, where a kinase protein, PKS5, is involved in the signaling and regulates the H^+^ATPase activity [80].

When extra CaSO_4_ (5 mM) was added to the cultivation medium of wheat cultivars with different salt tolerances, the cytosolic Ca^2+^ increased in both cultivars and decreased the overall concentration of Na^+^. On the other hand, the cytosolic pH increased only in the tolerant cultivar and the cytosolic Na^+^ increased only in the sensitive cultivar. It is likely that both higher cytosolic [Ca^2+^] and pH reflect tolerance mechanisms [81].

In experiments with field beans cultivated under salinity, both extra K_2_SO_4_ (10 mM) and extra CaSO_4_ (5 mM) caused similar increases of FW and DW [82]. Analysis using Oregon Green dextran dye and ratio imaging of the leaves demonstrated that extra calcium in vivo under cultivation acidified pH_apo_, but extra potassium had no effect, neither after 7 d nor after 21 d. After 21 d, both extra calcium and extra potassium reduced the cytosolic Na^+^, but after 7 d only calcium reduced the Na^+^ concentration. These changes were correlated with changes in the PM H^+^-ATPase hydrolyzing activity. Extra calcium addition induced an activation of the PM H^+^ATPase that resulted in a higher cytosolic pH.

It was concluded that for wheat and field beans, extra Ca^2+^ is better than K^+^ to decrease [Na^+^cyt], but for cultivation during a longer time, 21 d, even extra K^+^ is advantageous. Potassium treatment in vitro increases the H^+^-ATPase activity by binding to the phosphorylation domain of the ATPase. This reaction causes a dephosphorylation, which leads to the activation of the proton pump [306]. Moreover, both K^+^ and Ca^2+^ might bind to pockets at the phosphorylation and nucleotide-binding domains of the H^+^-ATPase and modulate its activity [307].

Kinetic studies of ATPase activity in field beans showed that the addition of extra calcium did not affect the apparent Km, but increased the Vmax of the H^+^-ATPase in a noncompetitive way both after 7 d and 21 d with and without salinity [82]. The increase in Vmax by calcium supply is likely to depend on the fact that more enzyme units are expressed. It is well known that calcium has an important role in regulating gene expression and improving transcriptional level of proteins under stress [308]. However, without extra calcium, salinity decreased the Vmax both after 7 d and 21 d. Extra potassium supply diminished both the apparent Km and Vmax in controls without salinity. Only after 21 d with salinity did K^+^ increase the Vmax and apparent Km [82]. The competition between K^+^ and Na^+^ may depend on the fact that protons are used both for Na^+^/H^+^ antiport and K^+^+H^+^ symport. The increase in Km suggests that the affinity between enzyme and substrate of the H^+^ATPase, MgATP, decreases. Probably, less competition exists.

These results explain why extra calcium during the cultivation of field bean will improve the growth of shoots and leaves. The increase in ATPase activity caused the acidification of the apoplast, alkalization of the cytosol, and a lower Na^+^_cyt_ level in the leaves.

## 8. Systemic Ca^2+^ Signaling

### 8.1. Electrical Signals Were First Proposed in Mimosa pudica

A long time ago, it was demonstrated that plants possess a system for the long-distance transmission of stimuli, such as wounding and other types of stress. When the plant *Mimosa pudica* was subjected to a drop of cold water, less than 10 °C, an active current was obtained that propagated through the stem, petioles, pinnae, and pulvini [309]. The rate of the systemic signal was dependent on temperature. It was also demonstrated that wounding triggered a signal that could pass through dead tissue and indicated that the signal was passively transported in the xylem by an apoplastic messenger, called a “Ricca factor” [309,310]. Later research has accentuated a role for electric and hydraulic signals together with Ca^2+^ and ROS in systemic signaling at wounding and high-light stresses [311,312].

### 8.2. Translocation of Amino Acids by Diffusion and Bulk Flow Activates Ca^2+^ Channels

By the use of voltage- and ion-selective microprobes, the authors of [313] could show that a mild salt solution (30–50 mM KCl) or amino acids (1 mM glutamic acid or 5 mM GABA) additions to the leaves of barley induced action potentials (APs), rapid voltage changes, that propagated basipetally and acropetally from leaf to leaf and from root to leaf. Salt additions caused biphasic voltage changes at a speed 20–30 cm/min, starting with a strong depolarization causing Ca^2+^ influx, and then anion efflux. The Ca^2+^-influx was reflected by a small depolarization, but the anion efflux caused a voltage break-through. A K^+^-efflux started after the depolarizing voltage had passed the K^+^ equilibrium potential.

Amino acid addition did not induce APs through depolarization, instead it was suggested that APs was caused by binding the amino acids to receptors or ligand-gated Ca^2+^-conducting channels. Thereafter, Ca^2+^ activated an anion efflux.

Recent results show that long-distance Ca^2+^ waves induced at wounding and salt stresses are mediated by diffusion and the bulk flow of the amino-acid messenger glutamate [314]. The authors propose that stress triggers a release of amino acids that diffuse in the apoplast from the local-stimuli perception to a distal tissue. During their translocation, they activate glutamate receptor-like channels, GLR3.3 and GLR3.6, which enable Ca^2+^ transport by bulk flow in the vascular tissue. An analysis made by quantitative imaging of the wild type and mutants expressing Ca^2+^ and glutamate fluorescent tracers shows that salinity and several other stimuli might induce Ca^2+^ waves with similar dynamics. In this case, the APs were connected with transient apoplastic pH increases (1 unit) and cytoplasmic pH decrease (0.5 units).

### 8.3. Electrical Signals and Glutamate May Be Involved

An investigation on systemic signaling in *Arabidopsis* triggered by wounding also indicates the involvement of electrical signals and Ca^2+^ [315]. The signals were dependent on glutamate and pH-regulated Ca^2+^ channels. Both the glutamate receptor-like proteins, GLR3.3 and GLR3.6, had a role in the signal transmission from leaf to leaf. The same proteins were involved in Ca^2+^ and electrical signaling from root to shoot, but the process required an inactivated proton pump AHA1. The later systemic signaling was also triggered by glutamate in the absence of stress. Moreover, the findings showed that a leakage of glutamate from the phloem into the apoplast occurred, as well as an increase in apoplastic pH, leading to an activation of the GLRs and systemic signaling.

### 8.4. Ca^2+^-ROS Interactions

Propagation of stress-induced signals in barley leaves was proposed to be mediated by different regulators, such as by changes in membrane potentials and ion fluxes [313] and by the involvement of reactive oxygen species, ROS, as suggested [316]. Choi et al. [184] reported that salt stress induced systemic cytosolic Ca^2+^ waves that were propagated within the root cortical and endodermal cells at a speed of approximately 400 µM/s.

Evans and co-workers [317] emphasized that salt stress triggers Ca^2+^ waves propagating through the whole plant, assisted by ROS. From experiments with *Arabidopsis*, the authors proposed that Ca^2+^ was transported from the vacuole into the cytosol by two-pore channels, AtTPC1s. The plasma membrane NADPH oxidase AtRBOHD can be activated by a cytosolic Ca^2+^ increase. The authors proposed a self-propagating Ca^2+^ --> ROS --> Ca^2+^ model, as the salt-induced cytosolic Ca^2+^ increase can activate the AtRBOHD to release ROS by phosphorylation and Ca^2+^ binding to its E-hand motifs, and that ROS can then trigger Ca release into the cytosol by ROS-regulated plasma membrane channels. This model is plausible, since the inhibition of AtRBOHD, either by ascorbate or diphenyliodonium, by the inhibition of Ca^2+^ channels by La or Ruthenium Red, or by knockout of the TPC1 gene, very much reduced the wave transmission speed that was obtained under the control measurements. As expected, overexpression of the gene led to an increased speed.

## 9. Conclusions

Salt-tolerant and sensitive species or cultivars exhibit different mechanisms for the influx and efflux of Na^+^, K^+^, and Cl^−^ and pH changes under salt stress. Most of the cultivated cereals like wheat, rice, and barley are sensitive to salinity. Halophytes and other salt-tolerant plants display a rapid transient uptake of Na^+^ into the cytosol, to prevent toxic Na^+^, while sensitive plants take up and retain Na^+^ for a longer time in the cytosol [56]. The SOS1 antiporter in the plasma membrane is important for salt tolerance as it transports Na^+^ from the cytosol into the apoplast or out from the cell. Recent findings suggest that CCX1, a Ca^2+^-cation exchanger in the plasma membrane, also inhibits excessive Na^+^ and ROS concentrations [71]. The Na^+^, K^+^/H^+^ antiporter NHX in the tonoplast transports both Na^+^ and K^+^ into the vacuole and is involved in pH regulation [68]. The efflux route for Na^+^ from the cytosol depends on the species and H^+^ATPase activities at each membrane. Some plants can prevent long-distance Na^+^ transport from root to shoot as high Na^+^ is very harmful for photosynthesis and this transport can be regulated at the xylem/parenchyma cell border, where Na^+^ is either transported into the xylem for further transport by the transpiration stream or is recirculated into the root.

Ca^2+^ is established as an important second messenger for abiotic and biotic stresses and for growth and development [258,291]. Ca^2+^ changes are often connected with pH changes, and pH is considered as a secondary messenger under salt stress too. K^+^ takes part in signaling together with Ca^2+^ in the CBL-CIPK network that leads to ion and pH homeostasis and ROS signaling [266]. Salt stress induces a cytosolic Ca^2+^ increase, which is sensed by the SOS, Salt Overly ystem, and signals are transmitted to the CBL/CIPK system [286,318] (Figure 7).

For a long time, researchers have tried to find the sensor for salt, probably localized either in the plasma membrane or in the cytosol. Some researchers suggested that the long C-terminal tail of the SOS1 protein could be the sensor [57,318]. It was recently proposed that the PM sphingolipid GIPC acts as a Na^+^ sensor, and that the Na-GIPC complex might activate an unknown Ca^2+^ channel [11]. However, in the suggested model [286], Na^+^ is bound to GIPC and Ca^2+^ is transported by the same lipid. In Figure 6, it is instead suggested that Na^+^ both binds to GIPC and is transported by this lipid, and that another channel transports Ca^2+^ across the membrane. The reason for this is that an earlier investigation with salt-tolerant rice shows that Na^+^ enters the cytosol before Ca^2+^ elevation is induced [287].

Research on Ca^2+^ signaling has focused on cellular cytosolic signaling, but Ca^2+^ signals travel within the whole plant. Rapid communication from cell to cell might occur by ROS producing the NADPH oxidase RBOHD, by the two-pore ion channel TPC1 or by the glutamate receptor-like channels GLR3.3 and GLR3.6 [311]. Recent results suggest that glutamate is a messenger involved in systemic Ca^2+^ signaling in plants [314]. Translocation by glutamate and other amino acids is supposed to activate glutamate-receptor-like channels for Ca^2+^, causing Ca^2+^ waves from the local stimulus to distal tissues and cells under salinity and wounding.

The new results concerning systemic Ca^2+^ signaling with glutamate as a messenger, and that GIPC might be a sensor for salt, are interesting and have improved our knowledge on salt-stress signaling. However, there is a gap in the knowledge between local cytosolic signals and systemic signaling waves in the whole plant. There are still many questions to answer. Is GIPC the only sensor of Na^+^, or are many different sensors at work? Little is known about the role of glutamate and its receptor proteins in different cells, tissues, and organs and if the Ca^2+^ waves show the same dynamics within the whole plant. How the systemic Ca^2+^ waves are connected with organellar signals is still enigmatic and needs further investigation.

## Figures and Tables

**Figure 1 plants-13-00046-f001:**
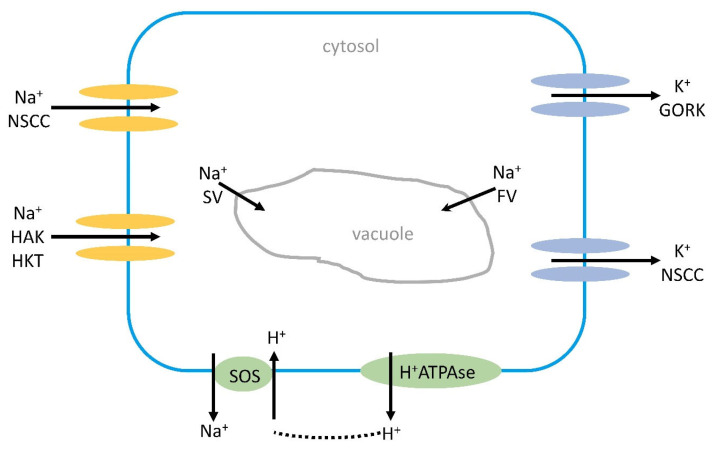
Na^+^ influx into a root cell or mesophyll cell in the leaf causes an efflux of K^+^ from the cell by GORKs or NSCCs by Na^+^-induced membrane depolarization. The depolarization might activate the H^+^ATPase in the plasma membrane, pumping out protons that can be used for the Na^+^/H^+^ antiporter (SOS1). Na^+^ can also leak into the vacuole by SVs or FVs. Under high salinity stress, most Na^+^ is transported via NSCCs and HKTs into cells, and also by HAKs. GORK, outwards-rectifying K^+^ channel; HAK, high-affinity K^+^ channel; NSCC, nonselective cation channel; SV, slow vacuolar channel; FV, fast vacuolar channel.

**Figure 2 plants-13-00046-f002:**
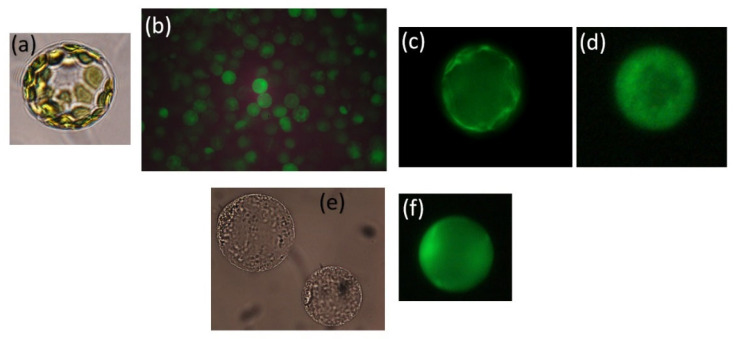
Protoplasts from rice mesophyll in transmitted light (**a**), and labelled with Fura 2 (**b**), labelled with SBFI (**c**), protoplasts from rice root labelled with SBFI (**d**), from wheat root in transmitted light (**e**), and labelled with SBFI (**f**). Fluorescence emission was measured at 530–550 nm.

**Figure 3 plants-13-00046-f003:**
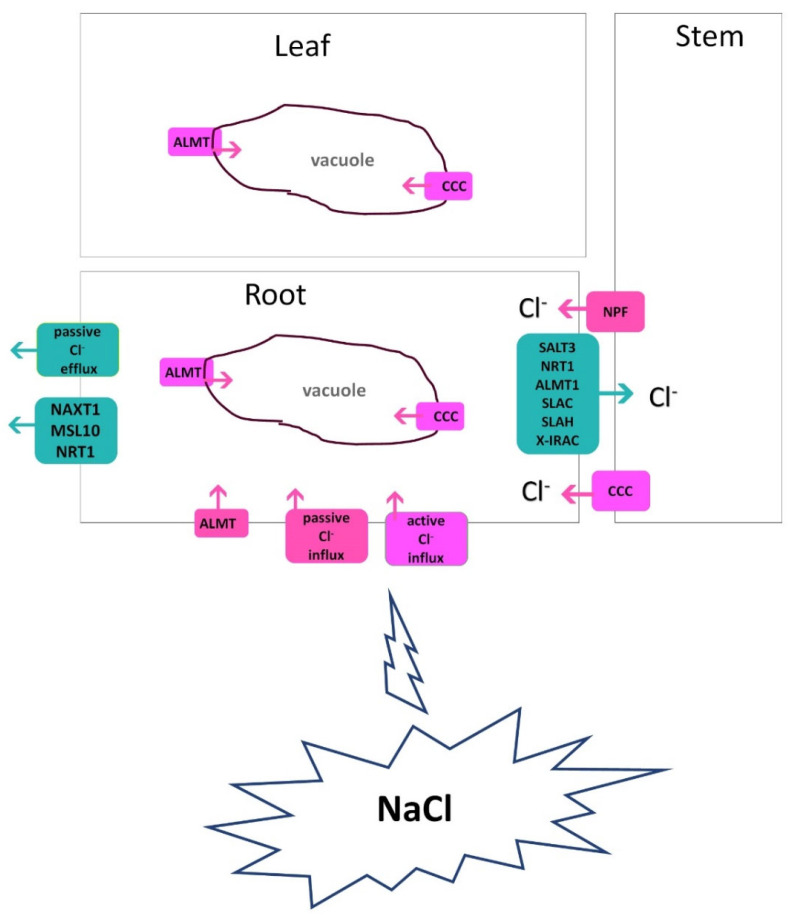
Chloride transport in the root, leaf, and stem of a plant via channels and transporters. SLAH, anion-channel-associated homolog; ALMT, aluminum-activated malate transporter; CCC, cation/chloride cotransporter; CLC, chloride channels; NRT, nitrate transporter; NPF, nitrate transporter 1/peptide transporter; SALT3, salt-tolerance-associated gene on chromosome 3; SLAC, slow anion channel; NAXT1, nitrate excretion transporter1, MSL10, mechano-sensitive ion channel 10; X-IRAC, inwardly-rectifying anion channel. For Cl^−^, various transporters, including NRT, NPF, SLAH, ALMT, and CCC, are involved in uptake and transport over long distances. Cl^−^ influx in the vacuole also involves ALMT and CLC.

**Figure 4 plants-13-00046-f004:**
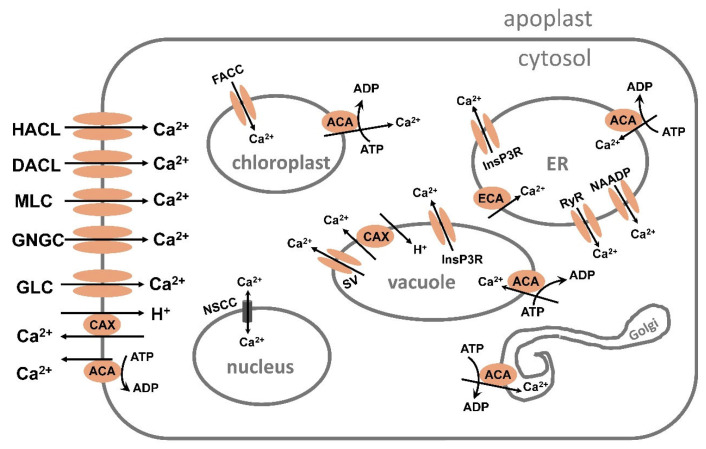
Ca^2+^ transport by channels, antiporters, and ATPase-mediated pumps in a plant cell. ACA, autoinhibited Ca^2+^ATPase; CAX, Ca^2+^ exchanger; CNGC, cyclic-nucleotide-gated channel; DACC, depolarization-activated cation channel; ECA, ER-type Ca^2+^ATPase; FACC, fast-activating cation channel; GLR, glutamate-receptor-like channel; HACC, hyperpolarization-activated cation channel; InsP_3_R, inositol 1,4,5-trisphosphate receptor-like channel; MLC, mechanosensitive-like channel; NSCC, nonselective cation channel; RyR, cyclic ADP-ribose (cADPR)-activator ryanodine receptor-like channel; SV, slow-activating vacuolar channel.

**Figure 5 plants-13-00046-f005:**
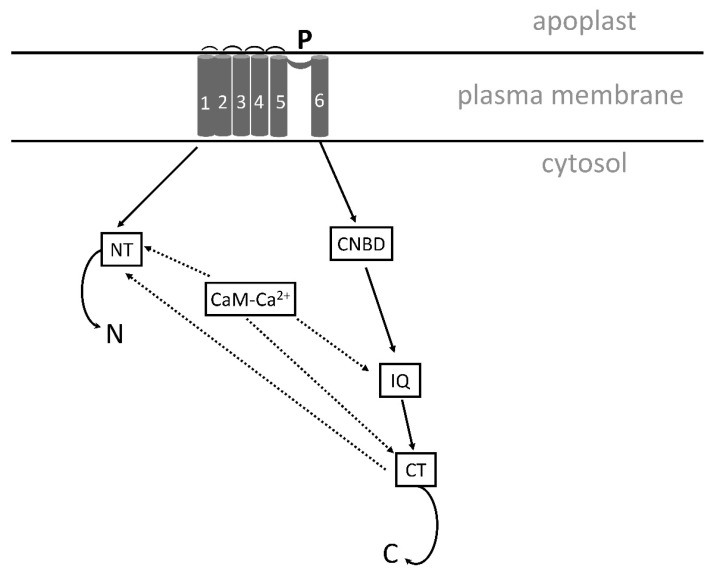
Simplified model of a cyclic nucleotide-gated channel (adapted from Demidchik et al. [269], with modification). The figure shows one subunit of the tetrameric channel with three CaM-Ca^2+^ (calmodulin-Ca^2+^)-binding sites. NT and CT, N- and C-terminal CaM-binding sites; IQ, isoleucine-glutamine domain, where CaM is bound; CNBD, cyclic-nucleotide-binding domain; P, part of the pore domain.

**Figure 6 plants-13-00046-f006:**
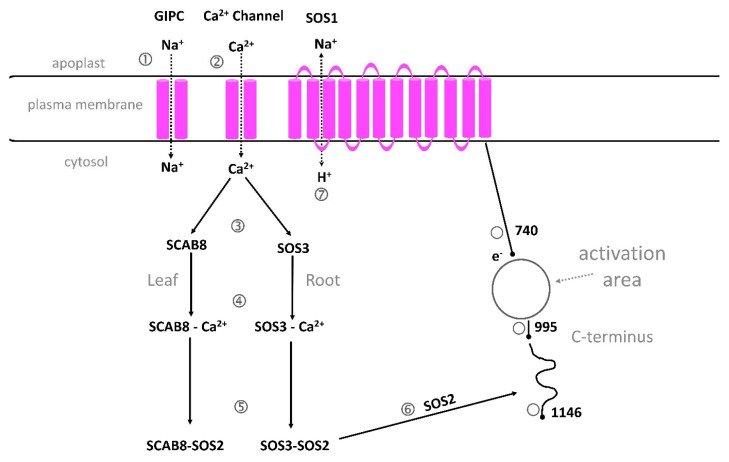
A possible model of SOS1 structure and its role in Na^+^-stress tolerance (adopted from Xie et al. [286]. SOS1, the salt overly sensitive 1 Na^+^/H^+^ antiporter, is composed of 12 transmembrane domains and transports Na^+^ out of the cell cytosol and protons into the cytosol. Na^+^ is proposed to bind to GIPC, a glycosyl inositol phosphoryl ceramide, a sphingolipid in the PM, and might also be transported into the cytosol by this lipid. The binding of Na^+^ to the lipid induces a transient efflux of Ca^2+^ into the cytosol, either via the sphingolipid molecule or by unknown Ca^2+^ channels. Ca^2+^ can be sensed by SOS3, a calcineurin B-like protein 10, and Ca^2+^-binding protein 8, SCABP8. These proteins can bind to and activate SOS2 and transport SOS2 to the serine phosphoryl domain 1136–1138 of SOS1 and activate this antiporter by phosphorylation: 749–925 activation area, 998–1146 C-terminal, 925–997 non-conservative area.

**Figure 7 plants-13-00046-f007:**
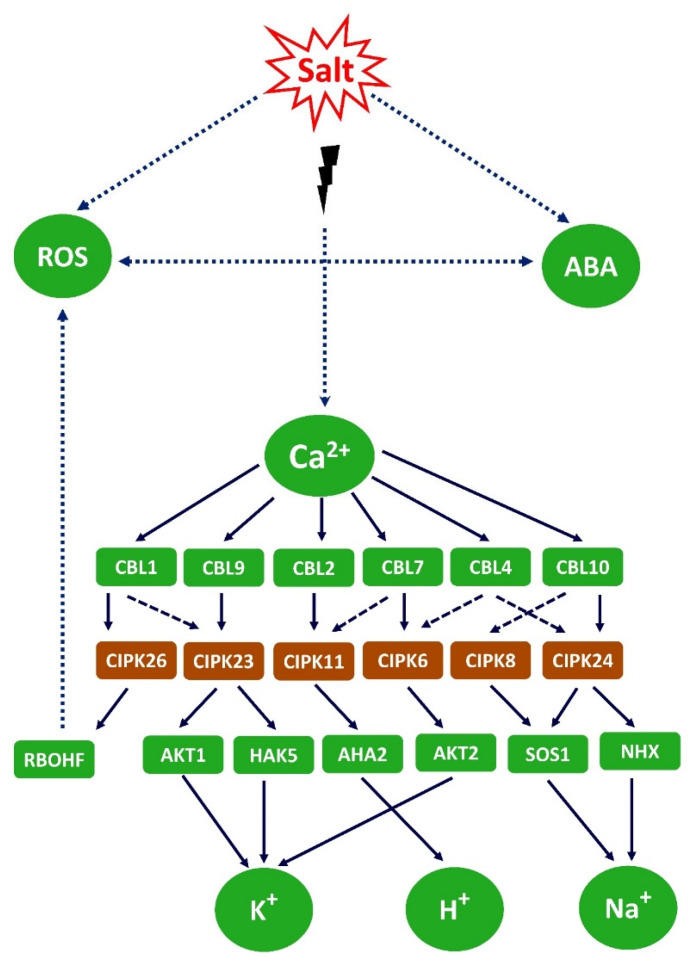
A simplified model of the CBL-CIPK network leading to ionic and pH homeostasis and ROS, reactive oxygen species, signaling. (i) Salt-stress-induced Ca^2+^ signals are sensed by CBL4 and CBL10, which combine with CIPK24 and CIPK8 to form CBL4/CIPK24 and CBL10/CIPK8, respectively, which activate the Na^+^/H^+^ antiporter in the PM and the Na^+^, K^+^/H^+^ antiporter in the tonoplast, respectively. This leads to less Na^+^ in the cytosol. (ii) ROS-induced Ca^2+^ signals are sensed by CBL1, which combines with CIPK26 and activates RBOHF, the respiratory burst oxidase homolog F, which produces ROS by Ca^2+^ binding to its EF-hand motif. (iii) K^+^ homeostasis is obtained by two mechanisms: In *Arabidopsis*, CBL1 combines with CIPK23 and activates AKT1 and HAK5, *Arabidopsis* K transporter1 and high-affinity K transporter, respectively, and CBL4 combines with CIPK6 and activates AKT2, K transporter 2. (iv) pH homeostasis (by H^+^) is obtained by binding between CBL7 and CIPK11, which activates AHA2, the PM H^+^ATPase, responsible for cytosolic pH homeostasis around 7.5 under normoxia.

## Data Availability

Not applicable.

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
