# Peer review of "Ion Changes and Signaling under Salt Stress in Wheat and Other Important Crops"

_plants, 2023, doi:10.3390/plants13010046_

Round 1
Reviewer 1 Report
Comments and Suggestions for Authors
This manuscript shows us how ions change and modify their signaling in wheat and other crops of economic importance subjected to salt stress. The manuscript is very well written, the information it gives us is very extensive and it is in my opinion the most complete document on ion homeostasis in plants subjected to salt stress. Both the text of the document and the figures clearly explain the effect of salinity on ionic transport and signaling in plant cells.
I consider that it is an important contribution on the subject, if I believe that despite all these virtues of the document, it is too extensive and I don't know if that could be an editorial problem. The English is very clear and is easy to understand for non-English speakers.
1. What is the main question addressed by the research?
The contribution of this review is that in a single document the changes in ion concentrations and their signaling in plants of interest are summarized. I consider that for those of us who work on salt stress it is of vital importance, not only for research work, but for students who carry out doctoral theses and master's degrees on the subject.
2. Do you consider the topic original or relevant in the field? Does it address a specific gap in the field?
The topic is original, since soil salinity is one of the most relevant problems that contribute to the loss of yields of glycophyt crops. The content of the review is a reference to address certain topics of interest in frontier research on salt stress. I believe that clearly showing us the processes involved in tolerance to salinity in different crops opens a door for the development of research aimed at this topic.
3. What does it add to the subject area compared with other published material?
The topic is current, today those who work in salt stress are eager for reviews that bring together the different phases of this type of stress. If we take this need into account, I would say that their most important contribution is, first, the point of view and the approach of the authors on the subject and secondly that it clearly summarizes each of the most important phases, it does not say much more than what is known today, but it says it in a different way, at least for our group it will be a reference to conceptualize future projects
4. What specific improvements should the authors consider regarding the methodology? What further controls should be considered?
Since this manuscript is a review, it does not have methodologies or statistics, therefore the topic is inappropriate.
5. Are the conclusions consistent with the evidence and arguments presented and do they address the main question posed?
I consider that the conclusions are appropriate to the topics discussed in the review and if it is a question, I believe that the fundamental question was related to the fact that the authors focused on how wheat and other important crops absorb Na, K and Cl when plants are under salt stress, as well as how calcium, potassium, and pH cause intracellular signaling and homeostasis. Similar mechanisms in the model plant Arabidopsis
6. Are the references appropriate?
The references are appropriate and 63% of them are from 2015 to 2023, which makes them a very good point for consultation, the review is very updated
7. Please include any additional comments on the tables and figures.
The figures are appropriate, they are very clear and are closely related to the text, the number is appropriate and the sizes too. There are no tables in the text although there may be some to buy any of the summarized mechanisms

Author Response
We appreciate and thank for your positive comments regarding our review very much!
Reviewer 2 Report
Comments and Suggestions for Authors
Dear Authors
Greetings!!!!
I have read your manuscript entitled “Ion changes and signaling under salt stress in wheat and other 2 important crops” with full interest and I found that the review article is very well written, by the authors, the overall quality of the review article is good. So, I do not have many major comments and your manuscript can be accepted in its present form
Author Response
We are very thankful for the comments on our review positively endorsing that our review can be accepted in the present form.
Reviewer 3 Report
Comments and Suggestions for Authors
Comments_2755431
The molecular mechanisms underpinning the tolerance and/or resistance of plants to salt stress is still elusive.
The authors have review the recent advancement done in this research field pointing the role of ion channels and ion carriers in the regulation of ion homeostasis in respond to a salt stress.
The manuscript is interesting but require some improvement (see comments).
Throughout the manuscript;
Use either the ion name or its chemical symbol.
Abstract
L 23. ‘Both input and efflux of Ca2+ from the cytosol affect … ‘
Reword the sentence, ‘…from the cytosol…' is correct for the efflux only!
Suggestion: ‘Both influx and efflux of Ca2+ affect....’
Main text
L 70. ‘The aquaporin-inhibitor H2O2 reduced Lp during the night, showing that these proteins were important for hydraulic conductivity. ‘
The conclusion must be done with caution.
1) Aquaporin facilitated the hydrogen peroxide (H2O2) diffusion through cellular membranes. 2) H2O2 is known to play a role in the redox signaling network, 3) H2O2 can directly or indirectly (ROS) damage biological membrane.
How can you exclude that the decrease in Lp is not a side effect link to ROS species?
L 74. ‘They are intrinsic protein channels in plasma membranes, ER, vacuoles and plastids’
There are ion channels in all cellular membranes. Why do you cite only four of them?
L 96. 2.3. Ion uptake across a membrane
This sentence must be deleted. It is of no interest for the understanding of the manuscript. One generally use equation to perform a quantitative analysis, which is not done in this manuscript.
Note that the conventional notation must be used when writing the electrochemical potential difference. For instance, delta must be replaced by its usual symbol. J is usually the symbol of the ion flux and not the activity. The symbol 'mu' is not defined.
might be the competition between K+ and Na+ at the uptake sites, as these ions have similar 203 ionic radius and ion hydration energy [50,51].
L 203. ‘might be the competition between K+ and Na+ at the uptake sites, as these ions have similar ionic radius and ion hydration energy [50,51].’
As far as I am aware, the ionic radius of potassium is larger than that of sodium. If the data are not given in the cited reference then remove them. Please be quantitative and write the values of ionic radius and hydrated ionic radius.
9. Conclusion
L1262. 'We have shown that…'.
Delete the beginning of the sentence. Use a direct and neutral style. ' Salt-tolerant and sensitive species or cultivars.....'
The conclusion contains seven times the word ‘important’. All active constituent involved in plant physiology might be thought as important. It would better to improve the style of conclusion to avoid this ‘important’ redundancy!
Minor comments
L 1112. 'changes [2295,296].'
one number must be corrected.
Author Response
Responses in a word doc is attached.

Reviewer 4 Report
Comments and Suggestions for Authors
The author addresses following suggestion
Line 12: Use Ca (calcium,) first time full explain the name
Line 26-29: add conclusion lines
So many minor mistakes in the abbreviation list, so I request to the author, please cross check all abbreviations.
Combine small paragraphs into one paragraph,, inside literature many small paragraph,
literature is enough for the review, just remove small typos
Author Response
Responses in a word doc is attached.

Round 2
Reviewer 3 Report
Comments and Suggestions for Authors
The manuscript has been improved and is acceptable for publication.